# EVA: GEOMETRIC INVERSE DESIGN FOR FAST MOTIF-SCAFFOLDING WITH COUPLED FLOW

**Yufei Huang**[1,2]**, Yunshu Liu**[2]**, Lirong Wu**[1,2]**, Haitao Lin**[1,2]**, Cheng Tan**[1,2]**, Odin Zhang**[1]**,
Zhangyang Gao**[1,2]**, Siyuan Li**[1,2]**, Zicheng Liu**[1,2]**, Yunfan Liu**[1,2]**, Tailin Wu**[2]**, Stan Z. Li**[2*]

[1] Zhejiang University, Hangzhou
[2] AI Lab, Research Center for Industries of the Future, Westlake University
`huangyufei@westlake.edu.cn; cairi@westlake.edu.cn`

## ABSTRACT

Motif-scaffolding is a fundamental component of protein design, which aims to construct the *scaffold* structure that stabilizes motifs conferring desired functions. Recent advances in generative models are promising for designing scaffolds, with two main approaches: training-based and sampling-based methods. Training-based methods are resource-heavy and slow, while training-free sampling-based methods are flexible but require numerous sampling steps and costly, unstable guidance. To speed up and improve sampling-based methods, we analyzed failure cases and found that errors stem from the trade-off between generation and reconstruction. Thus we proposed to exploit the spatial context and adjust the generative direction to be consistent with guidance to overcome this trade-off. Motivated by this, we formulate motif-scaffolding as a Geometric Inverse Design task inspired by the image inverse problem, and present *Evolution-ViA-reconstruction* (`EVA`), a novel sampling-based coupled flow framework on geometric manifolds, which starts with a pretrained flow-based generative model. `EVA` uses motif-aligned priors to leverage spatial contexts, guiding the generative process along a straighter probability path, with generative directions aligned with guidance in the early sampling steps. EVA is 70× faster than SOTA model RFDiffusion with competitive and even better performance on benchmark tests. Further experiments on real-world cases including vaccine design and multi-motif scaffolding demonstrate `EVA`'s superior efficiency and competitive performance.

## 1 INTRODUCTION

An important task in protein design is the generation of structural protein fragments, named *scaffold*, to support and stabilize a target *motif*. Here, motifs are structural protein fragments that carry the desired biological functions. Scaffolds, together with motifs, form complete, stable, and designable functional proteins. Motif-scaffolding achieves great success in applications to vaccine and enzyme design (Procko et al., 2014; Correia et al., 2014; Jiang et al., 2008; Siegel et al., 2010). It is similar to the task of in-painting or out-painting images (as illustrated in Fig. 1), which aims to generate high-fidelity and consistent images with the desired pixel patches (Chung et al., 2022; Song et al., 2022). What makes motif-scaffolding more complex is its 3D spatial and geometric nature, where we need to consider more factors (Castro et al., 2024; Wang et al., 2021), including spatial relationships, layout, physics, and so on to obtain realistic samples for real-world applications.

Generating diverse and designable scaffolds with accurate motifs of desire is highly challenging. With the development of techniques in protein structure generation, generative models such as Diffusion Probabilistic Models (DPM) have been successfully applied to motif-scaffolding (Trippe et al., 2023; Wu et al., 2023; Watson et al., 2023). Generative methods can be divided into two families: training-based and sampling-based. Training-based methods (Watson et al., 2023; Yim et al., 2024; Didi et al., 2023) train a motif-conditioned generative model that takes the motif directly as input and generates the remainder of the protein as the scaffold. These methods are **computationally intensive with slow inference speed** and inflexible to take advantage of different pretrained generative models,

---

[*]Corresponding Author

as they always need additional training. Sampling-based methods (Trippe et al., 2023; Wu et al., 2023) employ guidance (e.g., constrained gradients) in the sampling process of pre-trained generative models, bypassing the dependency on extra or specific training. These methods typically complete two processes at the same time. One is the reconstruction process, which ensures the accurate display of desired motifs with the aid of guidance. The other is the evolutionary (i.e., generative) process which helps to generate overall designable proteins from a broad distribution defined by pretrained generative models. Although training-free and flexible, sampling-based methods **require numerous diffusion sampling steps (500-1000 steps)** to gradually shift the generative path to the desired distribution, along with **computationally expensive and unstable guidance steps**, such as backpropagation or sequential Mont-Carlo steps. **The slow inference speed impedes the application of both training- and sampling-based methods**.

To further speed up and improve the sampling-based method, we investigate two typical types of failure for sampling-based methods: samples with low designability or inaccurate motif, as illustrated in Fig. 1. We attribute these errors to the inherent trade-off between generation (overall protein generation) and guidance (desired motif reconstruction) in sampling-based methods, which is also identified for similar image inverse problem solvers (Chung et al., 2022; 2024b). Thus, overcoming this trade-off and **adjusting the generative direction to be consistent with guidance** in earlier sampling stages are the keys to speeding up and improving success rates of sampling-based methods. To achieve this, we propose **leveraging the distinct spatial context** offered by the explicit point-cloud-like representations of proteins. Previous methods focused on applying sampling theories to the motif-scaffolding, with little exploration into the spatial contexts of protein point clouds with actual coordinates—such as the orientation and other geometric properties of desired motifs and generated structures—to better align the generative process with guidance and overcome the trade-off.

Motivated by the analysis above, we frame motif-scaffolding as a *Geometric Inverse Design* problem, providing a geometric perspective alongside the traditional posterior sampling view. We introduce *Evolution ViA reconstruction* (`EVA`), a fast, sampling-based method that uses a novel coupled flow framework on geometric manifolds, starting from a pretrained flow-based generative model. `EVA` uses motif-aligned priors to exploit spatial contexts and steers the generative process onto a straighter probability path, where the generative directions are aligned with guidance at early sampling steps. `EVA` is $70\times$ faster than current state-of-the-art RFDiffusion with competitive and even better performance. It can finish sampling in only 100 steps with nearly cost-free and highly stable guidance steps, which include simple spatial alignment and interpolation.

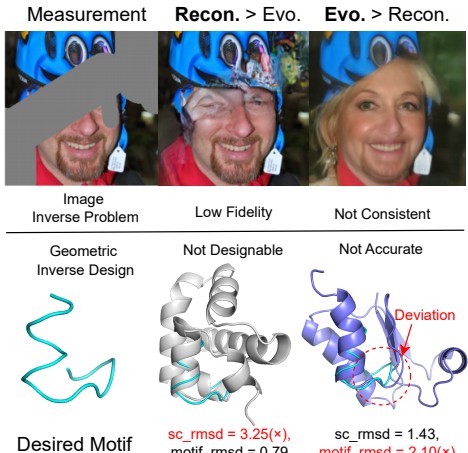

Figure 1: Two failures in the image inverse problem and geometric inverse design (motif-scaffolding). Recon. is short for motif reconstruction and Evo. is short for overall evolution (i.e, generation). Given partial observations, that is, measurement (*motif*), the image inverse problem (*motif-scaffolding*) aims to produce high-fidelity (*designable*) and consistent (*accurate*) complete images (*protein structures*).The images and proteins are generated with Chung et al. (2024b) and TDS (Wu et al., 2023).We interpret the ***'undes-ignability' issue*** as the *tangible **low-fidelity** issue* as they both seek for an overall harmonization.

In detail, we utilize flow-based methods pretrained with the flow matching objective (Lipman et al., 2022; Tong et al., 2024; Yim et al., 2023) for faster inference and more flexibility to control the sampling path. Instead of random noises, `EVA` starts with motif-aligned prior which leverages the spatial alignment for adjusting the global orientation and center of mass of initial point clouds to be aligned with given motif structure. This could reduce extra variance induced by a random alignment between motifs and prior point clouds, and avoid a twisted and unstable sampling process. At steps of sampling, `EVA` will estimate posterior means (i.e., predicted noise-free overall structures) that are consistent with desired motif via simple yet effective spatial interpolation of $C_\alpha$ coordinates and residue orientations. The estimated posterior will give a generative direction consistent with motif reconstruction.

To evaluate our method, we conducted experiments on the RFDiffusion dataset (Watson et al., 2023) and benchmarked EVA with various methods including state-of-the-art gradient correction methods (Pokle et al., 2024; Chung et al., 2022) extended to the geometric manifold. We also designed a new vaccine design benchmark that reflects real-world scenarios with our curated datasets. Furthermore, we test EVA in cases requiring to find optimal motif indexes or multi-motif relative positions. These conditions are provided in benchmark tests, rarely explored before, but actually not easily accessible in real-world cases. EVA shows comparative results in all benchmark tests with much reduced inference time. These results suggest our potential for realistic applications.

## 2  RELATED WORKS

**Motif-Scaffolding**  Wang et al. (2021) first proposed to use deep learning for motif-scaffolding. Recently, deep generative models (Trippe et al., 2023; Wu et al., 2023; Yim et al., 2024) have been applied for motif-scaffolding, which can be divided into two families: training-based and sampling-based. SMCDiff (Trippe et al., 2023) first proposed the sampling-based training-free method with diffusion and Sequential Monte Carlo(SMC). TDS (Wu et al., 2023) then improved upon SMCDiff using reconstruction guidance for each particle in SMC. These SMC-based method require much more network calls than directly sampling and thus are time-consuming. Wu et al. (2023); Yim et al. (2024) propose to use gradient correction for motif reconstruction. It is more efficient but requires numerous sampling steps and costly, unstable guidance. Training-based methods (Yim et al., 2024; Didi et al., 2023) like RFDiffusion (Watson et al., 2023) achieve state-of-the-art (SOTA) results on the motif-scaffolding benchmark. However, they rely on expensive fine-tuning of complex model architectures for conditional generation and have slow inference speeds. Our method lies in the sampling-based category with competitive performance and much improved efficiency, and thus has potential for various realistic applications.

**Image Inverse Problem**  Image inverse problem aims to recover the original image given some measurement (e.g. part of the image, noisy image). Recent methods solve this problem in a plug-and-play fashion via providing gradient correction by differentiating through the diffusion model in the form of reconstruction guidance (Ho et al., 2022), which is further extended in DPS (Chung et al., 2022) to nonlinear inverse problems. ΠGDM (Song et al., 2022) introduces pseudo-inverse guidance that improves the guidance approximation by inverting the measurement model. Pokle et al. (2024) extends gradient correction to the flow-based model. Gradient-based methods heavily rely on a deep gradient approximation of the intractable posterior score, which is costly to compute and crude for non-small noise levels at many steps of the diffusion process (Mardani et al., 2023; Chung et al., 2024a). General conditional generation methods can also be applied. SDEdit (Meng et al., 2022) is based on the replacement-based method which replaces the measurement-region of the generating samples with noisy given measurement. Our methods is as simple as replacement-based method with much better performance, which interpolates the predicted clean data with the clean given measurement. More related works can be found in the Appendix A.3.

## 3  BACKGROUNDS

### 3.1  FLOW MATCHING

In this section, we provide an overview of the general flow matching method to introduce the necessary notations and concepts based on (Pooladian et al., 2023; Tong et al., 2024; Lipman et al., 2022).

**Definition.**  Flow Matching is a family of simulation-free training objectives for continuous normalizing flow (CNF), which proposes to learn the time-dependent vector field $v(z, t) : \mathbb{R}^d \to \mathbb{R}^d$ to transform a sample $z_0 \in \mathbb{R}^d$ from an easy-to-sample prior distribution $p_0$ to a data point $z_1$ from the data distribution $p_1$. The vector field $v$ determines a unique time-varying flow $\psi_{t \in [0,1]} : \mathbb{R}^d \to \mathbb{R}^d$ through the Ordinary Differential Equation (ODE):

$$\frac{d\psi_t(z_0)}{dt} = v(\psi_t(z_0)) \quad \text{with boundary condition} \quad \psi_0(z_0) = z_0. \tag{1}$$

**Flow Matching.**  One wishes to find a vector field $v$ that pushes the flow $\psi_t$ to reach the desired data distribution $p_1$, i.e., $\psi_1 = p_1$. Generally, such a vector field $v$ is intractable, but can be learned by regressing the tractable conditional vector fields $u(z_t, t | z_0, z_1) = \frac{d}{dt} z_t$ where $z_t = \psi_t(z_0 | z_1)$ interpolates between endpoints $z_0 \sim p_0$ and $z_1 \sim p_1$. Theoretically, this interpolation defines the

time-dependent distributions $p(z_t|z_0, z_1)$, which are referred to as conditional probability paths. We focus on the probability path for the conditional Optimal Transport (OT) path (Lipman et al., 2022):

$$\psi_t(z_0|z_1) = z_t = (1-t)z_0 + tz_1, \quad p(z_t|z_0, z_1) = p(z_t|z_1) = \mathcal{N}(\alpha_t x_1, \sigma_t^2 \boldsymbol{I}) \tag{2}$$

where $\alpha_t = t$, and $\sigma_t = 1-t$. The conditional OT path has been demonstrated to have good empirical properties, including faster inference and better sampling in practice. Based on the conditional OT path, the conditional vector fields and Flow Matching (FM) regression objective are:

$$u(z_t, t|z_0, z_1) = u(z_t, t|z_1) = \frac{z_1 - z_t}{1 - t}, \quad \mathcal{L} = \mathbb{E}_{t, p_0(z_0), p_1(z_1)}\|\hat{v}(z_t, t) - u(z_t, t|z_1)\|^2 \tag{3}$$

where $\hat{v}(z_t, t)$ is a neural network to regress the conditional vector field. The optimal $\hat{v}$ that minimizes $\mathcal{L}_{FM}$ includes the posterior of conditional OT path and takes the form the below:

$$\hat{v}(z_t, t) = \frac{\hat{z}_1 - z_t}{1 - t}, \quad \hat{z}_1 = \mathbb{E}_{p(z_t|z_1)}[z_1|z_t]. \tag{4}$$

**Guidance-based conditional inference** The equation 4 above also holds with extra conditions $c$ (Pokle et al., 2024). The optimal vector fields for conditional inference could be obtained by replacing the unconditional posterior with conditional one $\mathbb{E}_p[z_1|z_t, c]$ in Eq. 4.

Given pretrained unconditional generative models that are trained to approximate $\mathbb{E}_p[z_1|z_t]$, guidance-based conditional inference methods typically convert the conditional posterior $\mathbb{E}_p[z_1|z_t, c]$ approximation into a guidance term approximation problem via Tweedie's identity (Yim et al., 2024; Chung et al., 2022; Song et al., 2022; Pokle et al., 2024). Applying this identity under Conditional OT path in Eq. 2 and simplifying gives (Pokle et al., 2024):

$$\mathbb{E}_p[z_1|z_t, c] = \mathbb{E}_p[z_1|z_t] + \frac{\sigma_t^2}{\alpha_t}\nabla_{z_t}\ln p(c|z_t),$$
$$\hat{v}(z_t, t, c) = \hat{v}(z_t, t) + \sigma_t\frac{\ln(\alpha_t/\sigma_t)}{dt}\nabla_{z_t}\ln p(c|z_t). \tag{5}$$

Thus, current guidance-based conditional inference methods focus on approximating the guidance term $\nabla_{z_t}\ln p(c|z_t)$ in a training-free manner. This forms the basis for various training-free conditional inference methods, including the state-of-the-art sampling-based methods for motif-scaffolding.

## 3.2 FLOW MATCHING ON THE PROTEIN GEOMETRIC MANIFOLD

**Notation.** The atom positions of each residue in a protein backbone structure could be represented as a residue frame tuple $g = (x, r)$ (Jumper et al., 2021b), where $x \in \mathbb{R}^3$ and $r \in SO(3)$ is the translation vector (i.e., the coordinate of the residue $C_\alpha$ atom) and rotation of a residue frame. The protein backbone consists of $N$ residues, thus it can be represented by $N$ residue frames as $\boldsymbol{g} = (g^{(1)}, ..., g^{(N)})$. We use bold face for collections of elements, superscripts for residue indices, and subscripts for time.

The protein geometric manifold $\mathcal{M}_P$ is a product space $\mathbb{G}$ of the 3D translation subspace $\mathbb{T}$ and the 3D rotation group $SO(3)^N$ of $N$ residues in the protein backbones. Here $\mathbb{T}$ is the Zero Center of Mass (CoM) subspace of $\mathbb{R}^{N \times 3}$, which means that the average of $N$ translation vectors should be zero to avoid the extra translation induced by the global rotation. As a start point, we first build the flow matching on this geometric manifold based on Riemannian flow matching (Chen & Lipman, 2023). It states that the time-dependent vector field in a manifold $\mathcal{M}$ can be defined as: $v(z, t) : \mathcal{M} \times \mathbb{R} \to \mathcal{T}_z\mathcal{M}$ where $t \in [0, 1]$ is the time step and $\mathcal{T}_z\mathcal{M}$ is the tangent space of the manifold at $z \in \mathcal{M}$. This vector field determines a CNF $\psi_{t \in [0,1]} : \mathcal{M} \to \mathcal{M}$. Since $\mathbb{G}$ is a product manifold, the tangent space is a direct sum: $\mathcal{T}_g\mathbb{G} = \mathcal{T}_x\mathbb{T} \oplus \mathcal{T}_r SO(3)^N \cong \mathbb{R}^{N \times 3} \oplus \mathbb{R}^{N \times 3}$ where $\boldsymbol{g} = (\boldsymbol{x}, \boldsymbol{r})$, and the flow ODEs proceed independently in the each manifold. Thus the construction of the flow on $\mathbb{G}$ is equivalent to building flows on each geometry independently. We can define the conditional vector field $u(\boldsymbol{g}_t, t|\boldsymbol{g}_1) = (u_x(\boldsymbol{x}_t, t|\boldsymbol{x}_1), u_r(\boldsymbol{r}_t, t|\boldsymbol{r}_1))$ of Riemannian flows on $\mathbb{G}$ with Conditional Optimal Transport path as follows (Chen & Lipman, 2023; Yim et al., 2023):

$$u_{\boldsymbol{x}}(\boldsymbol{x}_t, t|\boldsymbol{x}_1) = \frac{\boldsymbol{x}_1 - \boldsymbol{x}_t}{1 - t}, \qquad u_{\boldsymbol{r}}(\boldsymbol{r}_t, t|\boldsymbol{r}_1) = \frac{\log_{\boldsymbol{r}_t}(\boldsymbol{r}_1)}{1 - t}. \tag{6}$$

The first equation is straightforward. Since $\mathbb{T} \cong \mathbb{R}^{N \times 3}$, it is trivial to build the flow matching on translation in the same form as in Eq. 3. The second equation involves projecting elements in $SO(3)^N$ to its Lie algebra $\mathfrak{so}(3)^N$ using logarithmic mapping. Its core intuition is very similar to the case of $\mathbb{T}$, which computes an evolutionary direction from noisy data point $\boldsymbol{x}_t$ pointing to denoised data $\boldsymbol{x}_1$.

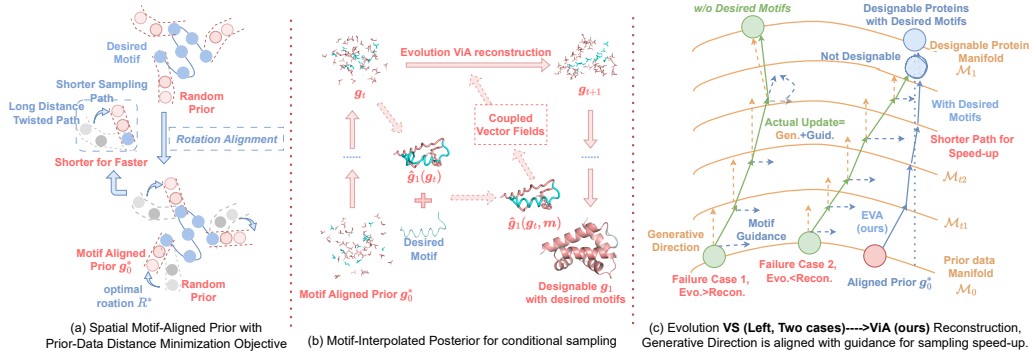

(a) Spatial Motif-Aligned Prior with Prior-Data Distance Minimization Objective    (b) Motif-Interpolated Posterior for conditional sampling    (c) Evolution **VS (Left, Two cases)----->ViA (ours)** Reconstruction, Generative Direction is aligned with guidance for sampling speed-up.

Figure 2: The illustration of `EVA` framework. Our key designs are twofold: (a) The random prior is first aligned with the desired motif through rotation (formally, the Eq. 8) to minimize the distance between them, resulting in a shorter sampling path and accelerated sampling. The aligned prior is referred to as the (Spatial) Motif Aligned Prior. (b) The conditional posteriors $\hat{g}(g_t, m)$ is approximated as the interpolation between unconditional posteriors and motifs $m$. This novel geometric solution could give generative directions consistent with motif reconstruction (formally, the coupled vector fields in Eq. 12) in early sampling stages.(c) Compared with gradient guidance-based methods, `EVA` could alleviate the conflict between generation and reconstruction (the failure cases 1 and 2, which require numerous sampling steps to gradually correct in gradient guidance-based methods) and thus need less sampling steps for better motif-scaffolding.

## 4 METHOD

This section describes the *Evolution ViA reconstruction* (`EVA`) framework. Our method draws inspiration from recent advances in flow matching (Song et al., 2023; Tong et al., 2024), which straighten flows during training for faster inference and better sampling in practice. We extend this intuition from training to conditional sampling for the speed-up of motif-scaffolding, which seeks to straighten the sampling path with given motifs. We achieve this through coupled flow, using a spatial motif-aligned prior that shortens the sampling path and motif-interpolated posterior, which aligns the generative direction with motif-reconstruction in earlier sampling stages. We summary our algorithm in Alg. 1 and a high-level schematic is provided in Fig. 2.

### 4.1 THE GEOMETRIC INVERSE DESIGN PROBLEM

We formulate motif-scaffolding as a Geometric Inverse Design problem. Formally, we observe motifs $g^M = \{g^{i_1}, ..., g^{i_k}\}$ of length $k$ where $\{i_1, ..., i_k\} \subset \{1, ..., N\}$ and the scaffold $g^S$ is all the remaining residues. Motifs meet the following conditions:

$$x^M = A x_1, \qquad r^M = \exp\left(A \log\left(r_1\right)\right). \tag{7}$$

where $g = (x_1, r_1) \in \mathbb{T} \times SO(3)^N$ is drawn from an unknown data distribution $p_1$, $N_m$ is the total length of motifs and $A \in \mathbb{R}^{k \times N}$ is a known or estimated motif mask matrix. The second equation involves converting between elements in $SO(3)^N$ and its Lie algebra $\mathfrak{so}(3)^N$ using Logarithmic and Exponential Mapping (Chen & Lipman, 2023; Bose et al., 2023; Yim et al., 2023).

Given a pretrained flow model with $\hat{v}(g_t, t)$, which is trained to approximate the unconditional posterior $\mathbb{E}_p[g_1 | g_t]$ as in Eq. 4 and the motifs $g^M$, **our goal is to generate designable protein backbones with accurate motif reconstruction via** *directly approximating the conditional posterior* $\mathbb{E}_p[g_1 | g_t, g^M]$ *in geometric space*.

Our perspective is different from previous training-based and sampling-based methods. Training-based methods (Watson et al., 2023) directly train a specific conditional model $v_\theta(g, t, g^M)$ that only samples the scaffold fragment while our formulation needs to sample the entire protein. As mentioned in Sec 3.1 and Eq. 5, the sampling-based methods use a proxy guidance gradient for conditional sampling while our formulation aims to directly approximate the conditional posterior $\mathbb{E}_p[g_1 | g_t, g^M]$ as a concrete protein point clouds using geometric methods. By leveraging the spatial context of explicit representations, our formulation offers a geometric solution to the generation vs. reconstruction trade-off, while the implicit approximation via guidance term is more exposed to the risk of the inherent trade-off.

## 4.2 THE GEOMETRIC SOLUTION

(Noisy) protein structures are not abstract latents or pixels but point clouds with specific coordinates and geometric properties (e.g., global orientations). The proposed EVA framework leverages these spatial contexts as a breakthrough point to overcome the trade-off and align the generative direction with motif reconstruction. Our geometric solution includes the **spatial motif-aligned prior** and **motif-interpolated posterior**,

The isotropic Gaussian prior used in the pretrained flow model may cause issues (Song et al., 2023) when directly applying the vector fields in Eq. 6. Since $\boldsymbol{x}_0$ (the point cloud from the prior) and $\boldsymbol{x}^M$ (the motif structure) are independent, random alignment between them introduces extra variance, leading to a twisted and unstable sampling process. To mitigate this issue, we introduce a mapping that aligns the global orientation and CoM between $\boldsymbol{x}^M$ and $\boldsymbol{x}_0$ to the 'closest' or optimal configuration, thereby reducing variance. The geometrically modified prior straightens the sampling path and paves the way for geometric interpolation as the orientations of data and prior are aligned. Additionally, noisy motif structure information provides an initial estimate for the motif parts of the prior. Further details can be referred in Sec. 4.3.

In Sec. 4.4, we introduce a geometric method for conditional sampling in the EVA framework. **Instead of complex models or guidance terms, we directly use geometric interpolation between the unconditional posterior $\mathbb{E}_p[\boldsymbol{g}_1|\boldsymbol{g}_t]$ and the target motif $\boldsymbol{g}^M$ to approximate the conditional posterior $\mathbb{E}_p[\boldsymbol{g}_1|\boldsymbol{g}_t, \boldsymbol{g}^M]$.** This gives us a clear evolutionary (i.e., generative) direction that aligns with motif reconstruction from the start by reconstructing the motif in the denoised protein $\hat{\boldsymbol{g}}_1$ (i.e., unconditional posterior), which we term *Evolution ViA reconstruction*. It is straightforward yet effective. Further analysis shows our approximation is equivariant to adding an extra OT conditional vector field corresponding to the motif reconstruction process. Thus, EVA steers the sampling process into a coupling of two conditional OT paths as in Eq. 6, which we call the ***Coupled Flow***.

## 4.3 SPATIAL MOTIF-ALIGNED PRIOR

Starting from prior $\boldsymbol{g}_0 = (\boldsymbol{x}_0, \boldsymbol{r}_0) \sim \mathcal{U}(\mathrm{SO}(3))^N \otimes \overline{\mathcal{N}}(0, I_3)^N$ following (Yim et al., 2023), where $\mathcal{U}(\mathrm{SO}(3))$ is uniform distribution over $SO(3)$ and $\overline{\mathcal{N}}(0, I_3)$ is the isotropic Gaussian with Zero CoM, we make it coupled with the motif from two aspects of orientation and specific coordinates.

**Motif Geometry-aware Prior** At the start of the sampling, we have access to the specific motif backbone coordinates $\boldsymbol{x}^M$ and the prior point clouds coordinates $\boldsymbol{x}_0$, we seek to estimate a mapping that aligns the global orientation and CoM between $\boldsymbol{x}^M$ and $\boldsymbol{x}_0^M$. Formally for the global orientation, we need to solve the equivariant optimal transport mapping as:

$$\pi^*, \boldsymbol{R}^* = \underset{\pi, \boldsymbol{R}}{\mathrm{argmin}} \|\pi(\boldsymbol{R}x_0^1, \boldsymbol{R}x_0^2, \ldots, \boldsymbol{R}x_0^N) - (x^{i_1}, x^{i_2}, \ldots, x^{i_k})\|_2. \tag{8}$$

where $\pi$ is a selection of $k$ residues as the motif in Eq. 7 out of $N$ total residues (i.e, defining the motif placement or indexes in the overall protein) and $\boldsymbol{R} \in \mathbb{R}^{3\times3}$ represents a rotation matrix in the 3D space. With both sides in the zero mass space (by subtracting CoM at first), the mappings in Eq. 8 are optimal for $E(3)$-equivariant transformations on either side of the point clouds (Song et al., 2023). Thus, we refer to these mappings as equivariant optimal transport (EOT).

The equivariant optimal transport finds the minimum straight-line distance between the paired atom coordinates upon all the possible rotations and alignment. Applying these mapping, we can obtain the optimized prior $\tilde{\boldsymbol{g}}_0 = (\tilde{\boldsymbol{x}}_0, \tilde{\boldsymbol{r}}_0) = (\boldsymbol{R}^*\boldsymbol{x}_0, \boldsymbol{R}^*\boldsymbol{r}_0)$, with the selected motif indexes $\pi^*$. For solving this EOT, we first enumerate the motif region using sliding windows on the entire protein sequences and then conduct Kabsch algorithm (Kabsch, 1976) to solve the optimal rotation matrix based on the atom alignments. We will try different motif regions in different samples of one target to finally obtain designable proteins. For multi-motif with unknown relative positions, we extend EOT in the prior to the generative process. More details can be referred to the Appendix C.1.

**Geometric Interpolation** Given the known motif structure $\boldsymbol{g}^M = (\boldsymbol{x}^M, \boldsymbol{r}^M)$, we can initialize the motif part (the other is still original prior) of the protein with noised motif structure at some time steps $t$ along the Gaussian probability path which interpolates the prior and the data, we note:

$$\tilde{\boldsymbol{x}}_0^M = (1-t) \cdot \tilde{\boldsymbol{x}}_0^M + t \cdot \boldsymbol{x}^M, \qquad \tilde{\boldsymbol{r}}_0^M = \exp_{\tilde{\boldsymbol{r}}_0^M}(t \cdot \log_{\tilde{\boldsymbol{r}}_0^M}(\boldsymbol{r}^M)). \tag{9}$$

This interpolation transforms the initialization of motif part into the conditional OT probability path of motif reconstruction at time $t$. This initialization $\boldsymbol{g}_0^*$ prepares for the subsequent coupling of the

overall structure conditional OT path and the motif reconstruction OT path, and also provides the prior with specific coordinate information of the desired motifs.

---

**Algorithm 1** A training-free Coupled Flow-based Framework for fast Motif-Scaffolding (`EVA`)

---

**Input**: Pretrained flow model $v_\theta(\boldsymbol{g}, t)$ which can estimate posterior $\hat{\boldsymbol{g}}_1(\boldsymbol{g}_t)$; Desired Motif Structure $\boldsymbol{m}$ as in Eq. 7; Enumerated or given (as in the benchmark test) motif region indexes $\pi^*$.
**Output**: Designable backbones $\boldsymbol{g}_1$ with desired motifs.

1: Sample: original prior $\boldsymbol{g}_0$, initial time $t_0$.
2: Obtain motif-aligned prior $\boldsymbol{g}_0^*$ via solving Eq. 8 with Kabsch algorithm.
3: **for** each time step $t \in [t_0, 1]$ of ODE integration **do**
4:     Get predicted unconditional posterior $\hat{\boldsymbol{g}}_1(\boldsymbol{g}_t) = (\hat{\boldsymbol{r}}_1, \hat{\boldsymbol{x}}_1)$ from pretrained $v_\theta(\boldsymbol{g}, t)$;
5:     Predict motif-interpolated conditional posterior as follow:
6:     $\mathbb{E}_p[\boldsymbol{r}_1^M|\boldsymbol{r}_t, \boldsymbol{r}^M] = \exp_{\hat{\boldsymbol{r}}_1^M}(\beta_t \cdot \log_{\hat{\boldsymbol{r}}_1^M}(\boldsymbol{r}^M)), \mathbb{E}_p[\boldsymbol{x}_1^M|\boldsymbol{x}_t, \boldsymbol{x}^M] = (1 - \beta_t) \cdot \hat{\boldsymbol{x}}_1^M + \beta_t \cdot \boldsymbol{x}^M$
7:     Calculating estimated conditional vector fields $\hat{v}_{\boldsymbol{x}}(\boldsymbol{x}_t, \boldsymbol{x}^M)$ and $\hat{v}_{\boldsymbol{r}}(\boldsymbol{r}_t, \boldsymbol{r}^M)$. as in Eq. 12;
8:     Continuing the ODE update step and get intermediate results $\boldsymbol{g}_t$;
9: **end for**
10: **return** Designable backbones $\boldsymbol{g}_1$ with desired motifs.

---

## 4.4 Evolution ViA reconstruction

Recall the vector field we defined in Eq. 6. For conditional sampling on the OT path as in Section 3.1, we can parameterize the conditional vector field as:

$$v_{\boldsymbol{x}}(\boldsymbol{x}_t|\boldsymbol{x}^M) = \frac{\hat{\boldsymbol{x}}_1(\boldsymbol{x}_t|\boldsymbol{x}^M) - \boldsymbol{x}_t}{1 - t}, \qquad v_{\boldsymbol{r}}(\boldsymbol{r}_t|\boldsymbol{x}^M) = \frac{\log_{\boldsymbol{r}_t}(\hat{\boldsymbol{r}}_1(\boldsymbol{r}_t|\boldsymbol{r}^M))}{1 - t}. \tag{10}$$

where $\hat{\boldsymbol{g}}_1 = (\hat{\boldsymbol{x}}_1, \hat{\boldsymbol{r}}_1)$ is the denoised prediction of $\boldsymbol{g}_t$. When $\hat{\boldsymbol{g}}_1(\boldsymbol{g}_t, \boldsymbol{g}^M) = \mathbb{E}_p[\boldsymbol{g}_1|\boldsymbol{g}_t, \boldsymbol{g}^M]$, the vector field $v(\boldsymbol{g}_t, \boldsymbol{g}^M)$ is the optimal, the same equations hold without $\boldsymbol{g}^M$ (Pokle et al., 2024).

We propose to explicitly approximate the conditional posterior $\mathbb{E}_p[\boldsymbol{g}_1|\boldsymbol{g}_t, \boldsymbol{g}^M]$ via geometric interpolation between unconditional posterior $\hat{\boldsymbol{g}}_1(\boldsymbol{g}_t)$ given by a pretrained unconditional flow model $\mathbb{E}_p[\boldsymbol{g}_1|\boldsymbol{g}_t]$ and $\boldsymbol{g}^M$, instead of implicit approximation via estimating the gradient of guidance and then combining gradients as in Eq. 5. Further analysis shows that this *spatial interpolation approximation* will lead to adding an extra vector field corresponding to a motif reconstruction OT path from the start. Thus, `EVA` steers the sampling process onto *a coupling of two OT paths* and estimates the *evolution* direction *via* directly reconstructing motif in $\hat{\boldsymbol{g}}_1$.

**Motif-interpolated Posterior** For simplicity, we denote the unconditional posterior $\hat{\boldsymbol{g}}_1(\boldsymbol{g}_t) = (\hat{\boldsymbol{x}}_1, \hat{\boldsymbol{r}}_1)$. During conditional inference ODE, the optimal vector fields $\hat{v}(\boldsymbol{g}_t|\boldsymbol{g}^M)$ are as follows:

$$\text{Conditional Inference ODE:} \quad \frac{d\psi_t(\boldsymbol{g}_0|\boldsymbol{g}^M)}{dt} = \hat{v}(\boldsymbol{g}_t, t|\boldsymbol{g}^M), \quad \psi_0(\boldsymbol{g}_0|\boldsymbol{g}^M) = \tilde{\boldsymbol{g}}_0, \tag{11}$$

$$\hat{v}_{\boldsymbol{x}}(\boldsymbol{x}_t|\boldsymbol{x}^M) = \frac{\mathbb{E}_p[\boldsymbol{x}_1|\boldsymbol{x}_t|\boldsymbol{x}^M] - \boldsymbol{x}_t}{1 - t}, \qquad \hat{v}_{\boldsymbol{r}}(\boldsymbol{r}_t, \boldsymbol{x}^M) = \frac{\log_{\boldsymbol{r}_t}(\mathbb{E}_p[\boldsymbol{r}_1|\boldsymbol{r}_t, \boldsymbol{r}^M])}{1 - t}. \tag{12}$$

We approximate the posterior $\mathbb{E}_p$ and obtain the estimated conditional vector fields as follows:

$$\mathbb{E}_p[\boldsymbol{r}_1^M|\boldsymbol{r}_t, \boldsymbol{r}^M] = \exp_{\hat{\boldsymbol{r}}_1^M}(\beta_t \cdot \log_{\hat{\boldsymbol{r}}_1^M}(\boldsymbol{r}^M)) \tag{13}$$

$$\mathbb{E}_p[\boldsymbol{x}_1^M|\boldsymbol{x}_t, \boldsymbol{x}^M] = (1 - \beta_t) \cdot \hat{\boldsymbol{x}}_1^M + \beta_t \cdot \boldsymbol{x}^M. \tag{14}$$

where $\beta_t = 1 - \gamma t \in [0, 1]$ is the coupling strength and $\gamma$ is an extra scaling factor that can be adjusted case by case. The scaffold region is still approximated with the unconditional posterior.

**Coupled Flow** Taking a closer look into its formula, we find the vector field is the coupling of the unconditional vector field and the motif reconstruction vector field, which are both corresponding to the conditional OT paths. Taking $\hat{v}_{\boldsymbol{x}}$ as example, inserting the expressions in Eq. 14 into Eq. 12 gives:

$$(1 - t)\hat{v}_{\boldsymbol{x}}(\boldsymbol{x}_t^M|\boldsymbol{x}^M) = (1 - \beta_t) \cdot \hat{\boldsymbol{x}}_1^M + \beta_t \cdot \boldsymbol{x}^M - \boldsymbol{x}_t^M$$

$$= (1 - \beta_t) \cdot \hat{\boldsymbol{x}}_1^M + \beta_t \cdot \boldsymbol{x}^M - \left[(1 - \beta_t) \cdot \boldsymbol{x}_t^M + \beta_t \cdot \boldsymbol{x}_t^M\right]$$

$$\hat{v}_{\boldsymbol{x}}(\boldsymbol{x}_t^M|\boldsymbol{x}^M) = (1 - \beta) \cdot v_{\boldsymbol{x}}(\boldsymbol{x}_t^M) + \beta \cdot v^M(\boldsymbol{x}_t^M).$$

| Method | Success Rate (%) | | | Designable Rate (%) | | | mRMSD %<1 | | | scRMSD %<2 | | | Div. | Solved | Time(s) |
|---|---|---|---|---|---|---|---|---|---|---|---|---|---|---|---|
| | <100 | >= 100 | All | <100 | >= 100 | All | <100 | >= 100 | All | <100 | >= 100 | All | | | |
| TDS | 43 | 13 | 36 | 88 | 85 | 87 | **48** | 13 | 40 | 56 | 43 | 52 | 161 | 19 | 63 |
| FF-G | 18 | 4.0 | 13 | 86 | 86 | 87 | 29 | 7.0 | 21 | 52 | 36 | 47 | 153 | 18 | 15 |
| RFDiff | 44 | **20** | **36** | 87 | **90** | **89** | 46 | **24** | **43** | 55 | 45 | 53 | 141 | **20** | 61 |
| **EVA** | **45** | 18 | **36** | **89** | 86 | 88 | 47 | 19 | 42 | **59** | 36 | 53 | **173** | **20** | **0.87** |

Table 1: Performance on the RFDiffusion benchmark of 24 targets. The best metrics are marked by **bold** and the second best metrics are marked by underline. <100 refers to targets with total length less than 100 residues and >= 100 refers to targets with total length greater or equal to 100 residues.

| **Method** | **Succ.** (%) | **Des.** (%) | **mRMSD** (%<1) | **scRMSD** (%<2) |
|---|---|---|---|---|
| SDEdit | 2.0 | 32 | 7.0 | 18 |
| DPS | 10 | 67 | 11 | 43 |
| FF-G | 12 | 85 | 16 | **46** |
| RFDiff | 16 | 91 | 19 | 45 |
| TDS | 14 | 83 | **22** | 22 |
| **EVA** | **17** | **94** | 18 | 41 |

Table 2: Performance on the Vaccine Design benchmark. The best metrics are **bold**. EVA can achieve better balance and thus higher success rate.

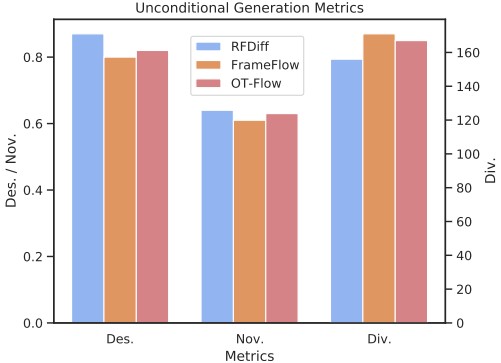

Figure 3: Unconditional Generation Results. The results suggest that both of the flow models can serve as strong unconditional foundations.

where $v^M(\boldsymbol{x}_t^M)$ is the vector fields determining a conditional OT path towards $\boldsymbol{x}^M$ for motif reconstruction and $v_{\boldsymbol{x}}(\boldsymbol{x}_t^M)$ is the original unconditional vector fields evolving on another conditional OT path for overall designable proteins. This coupling works on the motif regions. In early sampling steps, the motif reconstruction vector field quickly evolves the sampling motif towards the desired structure to align the overall generative direction with motif reconstruction, since the gradually reconstructed motif structure will influence the prediction of $\hat{\boldsymbol{x}}_1$. Later, the overall generative vector fields take over, enabling the designability of the entire protein. From the perspective of manipulating $\hat{\boldsymbol{x}}_1$, beyond motif-scaffolding, other conditional generation tasks that can formulate constraints on $\hat{\boldsymbol{x}}_1$ can utilize our framework. It simply requires modifying $\hat{\boldsymbol{x}}_1$ using gradient-based (simply on $\hat{\boldsymbol{x}}_1$ not deep gradients on $\boldsymbol{x}_t$) or direct modification methods and blending the modified with the original. We leave this for future work.

## 5 EXPERIMENTS

We justify the advantages of EVA with comprehensive experiments. The experimental setup is introduced in Section 5.1. As a start-up, we also report the unconditional generation performance for pretrained flow models. We aim to answer five research questions. **Q1:** How effective is EVA for motif-scaffolding against other training-free inverse problem solvers? **Q2:** How does EVA perform compared to the state-of-the-art motif-scaffolding models in efficiency and results **Q3:** Can EVA generalize well to more challenging and realistic tasks (e.g. vaccine design)? **Q4:** How do key designs impact the performance of **EVA**? **Q5:** what's the quality of generated samples of EVA?

### 5.1 EXPERIMENTAL SETUPS

**Datasets.** We conduct sampling for motif-scaffolding on the RFDiffusion Benchmark (Watson et al., 2023) of 24 targets following previous works (Trippe et al., 2023; Wu et al., 2023; Zheng et al., 2024). For vaccine design, we established an in silico benchmark test comprising 10 vaccine design targets addressed in recent publications, including epitopes from the respiratory syncytial virus (RSV) fusion protein (RSVF) that can produce neutralizing effect (Castro et al., 2024). More details can be referred in the Appendix B.

**Metrics and Sampling.** We use the self-consistency evaluation protocol following (Trippe et al., 2023; Yim et al., 2024; Zheng et al., 2024), including metrics as self-consistency backbone root-mean-square deviation (scRMSD), self-consistency motif rmsd (mRMSD), *Designability* (Des.), *Diversity* (Div.) and *Novelty* (Nov.). Given a generated protein sample, the self-consistency evaluation includes three stages: 1) inverse folding generated backbones with ProteinMPNN to get designed sequences. We predict 8 designed sequences per backbone following (Yim et al., 2024). 2) refolding the

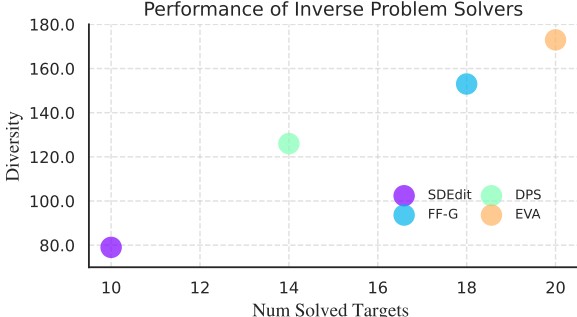

| Method | Succ. (%) | Des. (%) | mRMSD (%<1) | scRMSD (%<2) |
|--------|-----------|----------|-------------|--------------|
| FF-G   | 12        | 85       | 16          | **46**       |
| RFDiff | 27        | 85       | 27          | 45           |
| TDS    | 10        | **87**   | 13          | 25           |
| **EVA** | **28**   | 83       | **28**      | 38           |

Table 3: Performance on targets with 2 or more discontinuous motifs in the RFDiffusion Benchmark. The best metrics are marked by **bold**. EVA can achieve better balance on challenging targets and thus higher overall success rate.

Figure 4: Performance of inverse problem solvers on the RFDiffusion benchmark.

designed sequences via ESMFold. 3) calculating various metrics including RMSD, TM-score (Zhang & Skolnick, 2005), to evaluate the consistency (i.e., structural similarity) between the generated sample and the refolded sample. We use self-consistency TM-score(scTM) for evaluating the overall similarity between generated samples and refolded structures. scRMSD, which is more sensitive to minor structural variances, is also applied for a more stringent measurement. In addition, to judge whether the motif scaffolding was successful or not, we calculate the motif RMSD between the predicted design structure and the original input motif (mRMSD). A protein scaffold is *designable* if it meets: scTM >0.5, pLDDT >70 (pAE <5), which are confidence metrics employed in ESMFold or AlphaFold2 to ascertain the reliability of the self-consistency metrics. A designable scaffold with mRMSD <1 is considered a successful motif-scaffold.

We generate 100 scaffolds per target with 100 time-steps using Euler integrator, perform the self-consistency evaluation above on every scaffold, and report the percentage of samples that *designable* and successful (designable rate, *Des.* and success rate, *Succ.*), the percentage of samples with scRMSD <2 and mRMSD <1. Furthermore, we also report *diversity* in designable samples, which is the cluster number with TM-score Threshold set to 0.5. *Novelty* (Nov.) is also reported for the unconditional generation, which is the average TM-score of each sample to its closet protein in the PDB computed using FoldSeek (van Kempen et al., 2022). Average metrics across targets are reported. We evaluate the efficiency with the average time (over 100 samples in total for each method) for generating a 100 amino acid backbone in the same device.

**Baselines** We compare EVA with state-of-the-art motif-scaffolding methods including RFDiffusion (short as RFDiff) (Watson et al., 2023), TDS (Wu et al., 2023), and FrameFlow-Guidance (FF-G for short) (Yim et al., 2024) (and its pretrained flow model, FrameFlow (Yim et al., 2023)), and inverse problem solvers including SDEdit (Meng et al., 2022) (the Replacement method), DPS (Chung et al., 2022) (the Gradient Correction method), which we both extended to the geometric manifold. We use the original sampling set ups in their paper for fair comparison. **Unless otherwise specified, all conditional generation experiments are based on the pretrained FrameFlow (Yim et al., 2023) model.** In addition, we have reproduced the OT-Flow model (Bose et al., 2023) to test the flexibility of our approach. More details can be found in Appendix B. It is noted that we are not comparing the FrameFlow and OT-Flow. As long as they are successfully trained, we can use them for conditional sampling. We add OT-Flow as another choice just for the Ablation study.

## 5.2 RESULTS ON UNCONDITIONAL GENERATION

To demonstrate the flexibility of EVA, we additionally reproduced a protein flow model OT-Flow (Bose et al., 2023) and evaluated its performance. The unconditional generation result in Fig. 3 shows that we are using reliable pretrained generative models for training-free methods, which builds basic intuitions for the performance of the following conditional sampling. We evaluate these models by sampling 100 samples from lengths 70, 100, 200, 300 following (Watson et al., 2023).

## 5.3 RESULTS OF INVERSE SOLVERS (**Q1**)

Fig. 4 shows the results of different inverse problem solvers for motif-scaffolding. Our *EVA* significantly outperforms all inverse problem solvers including replacement-based and gradient correction methods, highlighting the better balance between reconstruction and evolution of EVA. The results imply when the unconditional generative bases are not as strong as in the image domains, training-free methods need to carefully reach a trade-off.

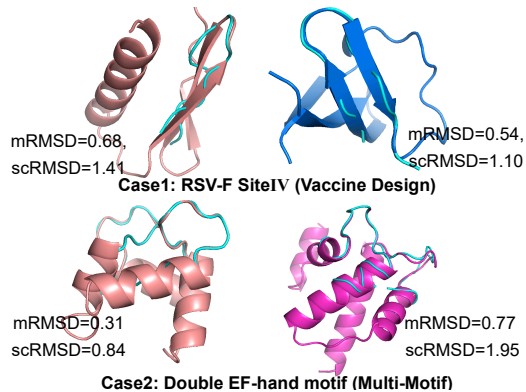

Figure 5: Case Study on RSVF4 and 1PRW.

| Method | Solved | mRMSD (%<1) | Des. (%) |
|---|---|---|---|
| **EVA** | **20** | **42** | 88 |
| EVA (OT-Flow) | 20 | 39 | 90 |
| DPS | 14 | 12 | 71 |
| w/o our Prior | 18 | 39 | 87 |
| Interp. to DPS | 14 | 15 | 69 |
| setting $\gamma = 1.1$ | 19 | 40 | **91** |
| setting $\gamma = 0.9$ | 20 | 41 | 86 |

Table 4: Ablation study for designed components on motif-scaffolding in RFDiffusion Benchmark. The best metrics are marked by **bold**. Des. is the short for Designable Rate.

## 5.4 RESULTS ON RFDIFFUSION BENCHMARK(**Q2**)

We compare EVA with state-of-the-art methods in RFDiffusion benchmark in Table 1. EVA is 70× faster than current state-of-the-art RFDiffusion, which supports our analysis about the trade-off and demonstrates the effectiveness of the spatial context. As a training-free approach, we achieve comparable performance with current SOTA training-based methods, RFDiffusion and superior performance on targets with total length <100. It should be noted that RFDiffusion is based on a much more complex protein structure prediction model, which is better at generating larger proteins and requires time-consuming structure prediction pretraining and generative fine-tuning. We also achieve leading performance across training-free methods, including SOTA particle-based SMC methods, TDS, which enjoys good performance in the cost of intensive guidance computation.

## 5.5 PERFORMANCE COMPARISON ON REALISTIC TASKS (**Q3**)

Vaccine design and multi-motif scaffolding are both important and challenging real-world tasks (Castro et al., 2024). We test EVA on those challenging tasks as in Table 2 and Table 3. Our methods has achieved competitive performance on the benchmark test, suggesting our scheme of directly approximating the posterior mean using spatial context is simple yet effective. We also tested cases requiring to find optimal motif indexes or multi-motif relative positions. More results and details can be referred in the Appendix C.1.

## 5.6 ABLATION STUDY (**Q4**)

As Table 4 suggests, components of EVA all contribute to our superior performance. The results highlight the flexibility of EVA, which could take advantage of different unconditional generative foundations. As guidance is the essential for motif-scaffolding, we replace the coupled approximation module within EVA with DPS guidance to study the importance of motif-aligned approximation.

## 5.7 CASE STUDY (**Q5**)

**EVA can generate designable scaffolds with accurate motif reconstruction.** Fig. 5 shows EVA can scaffold the epitope of RSVF-site4 and the double motif EF hand of Protein 1PRW successfully, suggesting the potential of our method for real-world tasks. More cases can be found in the Appendix. **EVA can generate diverse scaffolds** Fig. 5 shows different scaffolds with various lengths and secondary structure for two targets. Diversified scaffolds can improve the experimental success rate.

## 6 CONCLUSION

We formulate the motif-scaffolding as Geometric Inverse Design and identify its inherent trade-off between generation and motif reconstruction. Tailored to this perspective, we present EVA, a novel coupled flow framework on geometric manifolds, which uses motif-aligned priors and steers the generative process onto a straighter probability path, where the generative directions are aligned with guidance at early sampling steps. EVA is 70× faster than current state-of-the-art RFDiffusion with competitive and even better performance. The results on benchmarks demonstrate our efficiency and effectiveness. Limitations still exist, including insufficient exploration of general conditional sampling and motif layouts.

## 7 ACKNOWLEDGMENTS

Many thanks to Prof. Dapeng Li for helpful discussions and constructive feedback. This work was supported by National Science and Technology Major Project (No. 2022ZD0115101), National Natural Science Foundation of China Project (No. 624B2115, No. U21A20427), Project (No. WU2022A009) from the Center of Synthetic Biology and Integrated Bioengineering of Westlake University, Project (No. WU2023C019) from the Westlake University Industries of the Future Research Funding.

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

# A  BACKGROUNDS

## A.1  MORE ANALYSIS

A key challenge in motif-scaffolding lies in the trade-off between preserving accurate motifs and generating overall designable proteins (illustrated in Fig. 6. Traditional machine learning methods use time-consuming stochastic search techniques, taking hours to generate one scaffold (Wang et al., 2021). This hinders the creation of diverse scaffolds essential for experimental success.

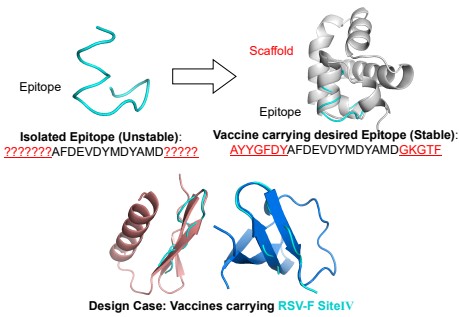

Figure 6: Cases requiring to find optimal motif indexes or multi-motif relative positions

Traditional machine learning-based methods rely on stochastic search techniques that require hours of computation to generate a single plausible scaffold, which is not desirable as sets of diverse scaffolds are needed to increase the likelihood of success in experimental validation.

Training or fine-tuning conditional generative models is a natural and effective idea. While these methods need lots of computing resources and a large number of protein motif-scaffold pairs for training, resulting in high computational costs. Moreover, training-based methods are **inflexible** to take advantage of different pretrained generative models and incorporate various practical constrains like spatial arrangements or layout for real-world applications, as they always need extra training or fine-tuning.

This limitation has its root in the Evolution Verse Reconstruction trade-off process mentioned before. The evolutionary process tries to sample from unconditional distribution to generate overall stable and designable proteins, while the guidance term focuses on the accurate reconstruction of desired motifs. Thus, the direction of Evolution and Reconstruction could be conflict. This often leads to two types of unsatisfied samples with low designablity or inaccurate motif integration (as illustrated in the Fig.1). The conflict demonstrated the fact that current conditional generative frameworks **fail to fully utilize motif information** to guide the evolutionary path of the generation process. Additionally, the geometric information. In addition, motif geometry information such as orientation can provide clues to the evolution of the overall protein, and existing methods **fail to explicitly account for these geometric constraints**.

we formulated the motif-scaffolding problem as the Geometric Inverse Design task and identified the Evolution Verse Reconstruction issue in this task. We introduce EVA with Evolution ViA reconstruction, a flexible and novel coupled flow matching framework on geometric manifolds, (which uses measurement-coupled priors and steers the generation process explicitly (vs implicitly as gradient) ) with the measurement-coupled Evolution ViA reconstruction plugin-and-play module, on E(3) Equivariant Optimal Transport Path. EVA can better balance the evolution and reconstruction process within generation and achieve comparable performance with greater flexibility.

## A.2  FURTHER COMPARISON WITH GRADIENT GUIDANCE

For ease of comparison, the following analysis will be conducted under the unified formulation of flow-based models.

In algorithmic sense, grad guidance includes two step: a. estimate the unconditional vector field and unconditional posterior $\hat{x}_1$ b. estimate the guidance gradient $\nabla_{x_t} \log p(x^M | x_t)$ and the composition

of the two:

$$\hat{v}(x_t, t|x^M) = \hat{v}(x_t, t) + \sigma_t \frac{\ln(\alpha_t/\sigma_t)}{dt} \nabla_{x_t} \log p(x^M|x_t),$$

$$\nabla_{x_t} \log p(x^M|x_t) = \nabla_{x_t} \|x^M - \hat{x}_1^M\|_2^2. \tag{15}$$

Where the guidance gradient is usually estimated by the assumption that $p(x^M|x_t) \approx p(x^M|\hat{x}_1^M)$ is a Guassian distribution. This is the origin of this term $\nabla_{x_t}\|x^M - \hat{x}_1^M\|_2^2$ .

**Our method differs from grad guidance in the second step**, given the estimated $\hat{x}_1$, we directly estimate the conditional posterior $\mathbb{E}_p[x_1|x_t, x^M]$, via interpolation between $\hat{x}_1^M$ and $x^M$. Then the conditional update direction is given by:

$$\mathbb{E}_p[x_1^M|x_t, x^M] = (1 - \beta_t) \cdot \hat{x}_1^M + \beta_t \cdot x^M,$$

$$\hat{v}(x_t, t|x^M) = \frac{\mathbb{E}_p[x_1|x_t, x^M] - x_t}{1 - t} \tag{16}$$

**Our method doesn't rely on any assumption (e.g. gaussian distribution assumption in FrameFlow-Guidance or Chroma) about the form of $p(x^M|x_t)$ to estimate its gradient.** Instead of two independently estimated gradients, our method gives the overall update direction directly.

Furthermore, our vector field for motif-scaffolding can be viewed as a composition of vector fields for overall generation and motif reconstruction:

$$\hat{v}_x(x_t^M|x^M) = (1 - \beta) \cdot v(x_t^M) + \beta \cdot v^M(x_t^M)$$

$$v^M(x_t^M) = \frac{x^M - x_t}{1 - t} \tag{17}$$

### A.3 MORE RELATED WORK

**Motif-Scaffolding** SMC-Diff and TDS utilize Sequential Monte Carlo (SMC) algorithm to re-purpose pretrained DPM for motif-conditioned sampling. Though training-free, these SMC-based methods need much more network calls to incorporate heuristic approximations, which may result in unsatisfied and slow generations. Gradient correction-based methods frame the motif-conditioned sampling as a problem of posterior sampling. Multi-motif scaffolding (Ke et al., 2024) was also explored, which treat the motifs as floating anchors. We refer a recent benchmark work (Zheng et al., 2024) for further reading.

**Inverse problem** Recently there are also many training-based methods to solve the inverse problem to seek for effectiveness. Notably, Diffusion Bridge-based method (Chung et al., 2023a) applies the similar idea of data-coupling as our methods. Improved gradient-correction method has also appeared (Chung et al., 2023b) which uses geometric decomposition.

**Protein Structure Prediction** The self-conditioning approach (Stark et al., 2023; Huang et al., 2023) is from the recycling mechanism in AlphaFold2 (Jumper et al., 2021a; Huang et al., 2024a) for improved structure prediction (Hu et al., 2023; Huang et al., 2024b; Hu et al., 2022). It takes the previous predicted posterior as model input and encodes the posterior to embedding for conditioning next iteration. In contrast, our method just takes the posterior for interpolation which will not be taken as model input for embedding.

## B EXPERIMENTS SETUP

**Inference set-up** All baselines and our approach are implemented using the PyTorch 1.6.0 library with Intel(R) Xeon(R)Gold6240R@2.40GHz CPU and NVIDIA A100 GPU. For a fair comparison, we follow the original inference set-up of all baselines (Yim et al., 2024; Wu et al., 2023; Watson et al., 2023). Previous sampling-based methods usually require 500 inference steps for reproducing their original performance, more than that in EVA. All sampling-based methods use a batch of 25 for every case, generating 100 scaffolds in total in a single A100 GPU. Because of the memory burden of running RFDiffusion, we instead use a batch of 2 in a single A100 GPU. The inference code framework is kept the same as TDS (Wu et al., 2023). The only difference for each sampling-based method is the time-step number and the conditional sampling implementation.

**Data Curation**   We download the RFDiffusion benchmark sets via PDB. The data list can be found in  (Watson et al., 2023). The vaccine dataset (Watson et al., 2023; Castro et al., 2024; Correia et al., 2014) include targets as follows: RSV Fusion Protein Site 0 (4JHW), Site 2 (3IXT), Site 4 (4ZYP), Site 5 (5TPN), RSV G-Protein 2D10 Site (5WN9); epitopes of DENV2 (pdb_id: 6FLA) that corresponding to neutralizing antibodies: 3E31, Z004, Z021, 3H5, 2C8, 2D73). We illustrated these neutralizing antibodies below in Figure 7:

## Neutralizing antibodies targeting DENV2 EDIII

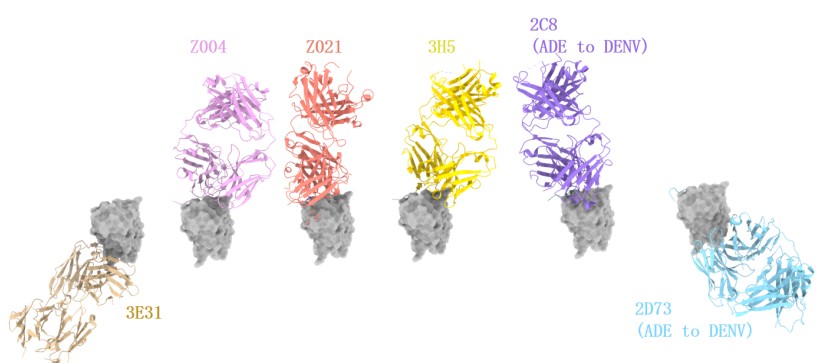

Figure 7: Neutralizing antibodies targeting DENV2 ED3

**Baselines Implementation**   We implement all the inverse solver baselines according to their official GitHub repositories and change the data structure from images to protein. The only modification is the normal distribution defined in different manifold ($\mathbb{R}^d$ to SO(3), E(3)). We follow the normal distribution definitions of TDS (Wu et al., 2023) and FrameFlow (Yim et al., 2023; 2024). The OT-Flow is reproduced based on the FoldFlow (Bose et al., 2023), using their official Code Repository for the SO(3) optimal transport path version. As we are not comparing the OT-Flow and FrameFlow, we can just use the default training settings in their original papers. As long as the pretrained flow model is capable of generating high-quality unconditional backbones (which we test in the Fig. 5), we can use them as generative foundations for training-free motif-scaffolding.

**Solving Equivariant Optimal Transport**   Our proposed algorithm solves Eq.7 approximately based on enumerated motif selections with sliding windows (i.e, enumerating the start indexes of motif segments) and kabsch algorithm for optimal rotation. The enumeration used here is for estimating multiple possible motif selections to increase the diversity of generated backbones, which contributes to the overall success rate. It's efficient since the sequential enumeration space is not big (total length - motif length or constrained by prior) and a single enumeration can determine motif selection for $N$ samples (by simply selecting the top $N$), allowing the computation time to be amortized across multiple samples. Kabsch algorithm is highly efficient and the runtime is negligible. Additionally, for long scaffolds with multi-motif and no prior constrains, we will apply a variant of iterative closest point algorithm (Song et al., 2023), where it iteratively obtains motif selection and Rotation.

## C   MORE RESULTS

We show the numerical results of unconditional generation and inverse problem solvers in Table 5 and Table 6. More case studies are introduced below.

### C.1   MULTI MOTIF SCAFFOLDING

As mentioned in the main text, we also test two cases requiring to find optimal motif indexes or multi-motif relative positions. These conditions are provided in benchmark tests, rarely explored before, but actually not easily accessible in real-world cases. As these are not our main focus, we just want to show the flexibility and extensibility of our method in a proof-of-concept manner. We

| Method | Des.($\uparrow$) | Div.($\uparrow$) | Nov.($\downarrow$) |
|---|---|---|---|
| RFDiff | **0.87** | 156 | 0.64 |
| FrameFlow | 0.80 | **171** | **0.61** |
| **OT-Flow** | 0.82 | 167 | 0.63 |

Table 5: Unconditional generation results.

| Method | Solved($\uparrow$) | Div($\uparrow$) |
|---|---|---|
| SDEdit | 10 | 79 |
| FF-G | 18 | 153 |
| DPS | 14 | 126 |
| **EVA** | **20** | **173** |

Table 6: Performance of inverse problem solvers on the RFDiffusion benchmark. The best metrics are marked by **bold**.

conduct the case study on protein 1PRW, we simulate the two scenarios by not providing the model with the motif index from existing cases or randomly rotating or translating the given motif structures. The results are show in Fig 8. For the case with unknown motif indexes, they involve multiple factors: 1) the spatial location of the motif, 2) the length of the motif itself, and 3) the distances between motifs. The final evaluation criterion is whether a designable backbone with the desired motif can be generated, and there is currently no numerical standard for this. Here, we present a heuristic enumeration method. We assume that the relative positions of multiple motifs are known

---

**Algorithm 2** A training-free Coupled Flow-based Framework for fast Motif-Scaffolding (EVA)

**Input**: Pretrained flow model $v_\theta(\boldsymbol{g}, t)$ which can estimate posterior $\hat{\boldsymbol{g}}_1(\boldsymbol{g}_t)$; Desired Motif Structure $\boldsymbol{m}$ as in Eq. 7; Enumerated or given (as in the benchmark test) motif region indexes $\pi^*$.
**Output**: Designable backbones $\boldsymbol{g}_1$ with desired motifs.

1: Sample: original prior $\boldsymbol{g}_0$, initial time $t_0 \in [0.1, 0.2]$.
2: Obtain spatial motif-aligned prior $\boldsymbol{g}_0^*$ with Kabsch algorithm.
3: **for** each time step $t \in [t_0, 1]$ of ODE integration **do**
4:     Get predicted unconditional posterior $\hat{\boldsymbol{g}_1}(\boldsymbol{g}_t) = (\hat{\boldsymbol{r}}_1, \hat{\boldsymbol{x}}_1)$ from pretrained $v_\theta(\boldsymbol{g}, t)$;
5:     Adjusting motif relative positions according to Eq. 18 and get updated $g_m$.
6:     Predict motif-interpolated conditional posterior as follows:
7:     $\mathbb{E}_p[\boldsymbol{r}_1^M | \boldsymbol{r}_t, \boldsymbol{r}^M] = \exp_{\hat{\boldsymbol{r}}_1^M}(\beta_t \cdot \log_{\hat{\boldsymbol{r}}_1^M}(\boldsymbol{r}^M)), \mathbb{E}_p[\boldsymbol{x}_1^M | \boldsymbol{x}_t, \boldsymbol{x}^M] = (1 - \beta_t) \cdot \hat{\boldsymbol{x}}_1^M + \beta_t \cdot \boldsymbol{x}^M$
8:     Calculating estimated conditional vector fields $\hat{v}_x(\boldsymbol{x}_t, \boldsymbol{x_m})$ and $\hat{v}_r(\boldsymbol{r}_t, \boldsymbol{x_m})$ as in Eq. 12;
9:     Run the ODE update step as in Eq. 12 and get intermediate results $\boldsymbol{g}_t$;
10: **end for**
11: **return** Designable backbones $\boldsymbol{g}_1$ with desired motifs.

---

(the case where the relative positions of motifs can vary independently will be discussed later). For a single motif, we only need to enumerate its starting index using a sliding window method, ensuring it remains within the protein index range. For multiple motifs, we first calculate the number of intervening amino acids based on the spatial distance between motifs plus their interaction radius (approximately 2-5 Å), divided by the average amino acid length. Since this value only simulates the scenario where amino acids are arranged in a straight line, it represents a minimum. We then enumerate the relative number of amino acids between each motif, i.e., the index difference between each motif. Once we have enumerated the index differences between motifs, the problem converts back to the single motif case, and we can continue using the sliding window for enumeration. For the case with unknown relative motif positions, we assume known motif indexes. Or it's a little complex and needs a slow iterative algorithm. The algorithm first conducts the Hungarian algorithm to align the atoms between the initial geometry from $p_0$ and the ground truth geometry from $q_1$; and then conducts the Kabsch algorithm to solve the optimal rotation matrix based on the atom alignment. The proposed algorithm asymptotically converges to the optimal solution. But it's slow for accurate exploration of the motif indexes and motif relative positions product space.

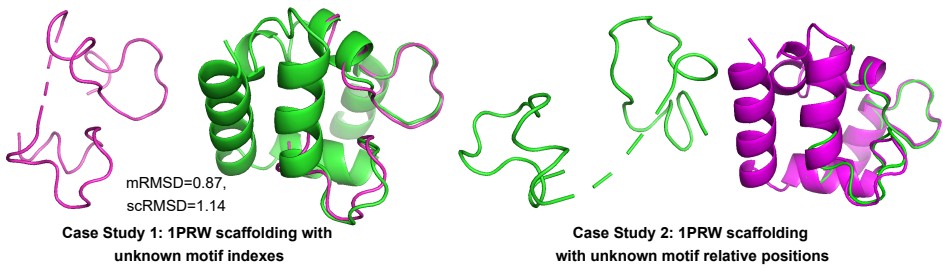

mRMSD=0.87,
scRMSD=1.14

**Case Study 1: 1PRW scaffolding with**       **Case Study 2: 1PRW scaffolding**
**unknown motif indexes**       **with unknown motif relative positions**

Figure 8: Cases requiring to find optimal motif indexes or multi-motif relative positions

On the other hand, if the motifs are independent, we can assume a relative far sequential distances between the motifs. This intuition comes from the fact that these motifs don't need to interact with each other and their relative positions are not necessarily to be close. So we just keep the independent motifs independent and assign them motif indexes that are far in sequential order. Note the independent motifs could still be spatially close although they are sequentially far. But it doesn't violate our assumptions of independent motifs. Therefore, we can simply enumerate some motif indexes that are relatively far apart in the sequence, and then focus on exploring their relative positions.

We can explore the motif relative positions with a modified version of EVA. This time, we need to adjust the EOT objective as in Eq. 18:

$$\pi^*, \boldsymbol{R}^* = \operatorname*{argmin}_{\pi, \boldsymbol{R}} \|\pi(x_t^1, x_t^2, \ldots, x_t^N) - (\boldsymbol{R}x_{\boldsymbol{m}}^1, \boldsymbol{R}x_{\boldsymbol{m}}^2, \ldots, \boldsymbol{R}x_{\boldsymbol{m}}^{N_m})\|_2 \tag{18}$$

In every step (or in just some selected steps for efficiency), we take the intermediate sampling result $x_t$ as a relative reference and use the Kabsch algorithm based on Eq. 18 to compute the optimal rotation for each motif relative to its corresponding prior part. Then, we apply the optimal rotations and perform Geometric Interpolation to calculate the conditional posterior. This method implicitly leverages the pretrained generative model to explore the relative positions of the motifs. As the protein structure gradually folds into a stable state during generation, the generative model makes spatial adjustments for each motif part. Our approach aligns with these adjustments step by step through spatial alignment, while also ensuring that the motif structure itself conforms to the given structure via Geometric Interpolation. We omit the trivial CoM alignment for simplicity. We summary the algorithm in Alg. 2.

## C.2 MORE ADVANCED RESULTS

**More ablations.** We provide the performance of FrameFlow against sampling steps in Table 7, more ablation study of hyperparameters of EVA in Table 8, and more case study results for the RFDiffusion benchmark in Table 9. The ablations suggest the hyperparameters could control the strength of motif reconstruction and the designability and users could adjust them case by case. We find simply setting $\gamma = 1, t_0 = 0.1$ is enough for most cases.

| Method | Solved | mRMSD(%<1) | Des(%) | Time(s) |
|---|---|---|---|---|
| FrameFlow-guidance (100 steps) | 16 | 14 | 79 | 3.1 |
| FrameFlow-guidance (300 steps) | 16 | 18 | 83 | 9.7 |
| FrameFlow-guidance (500 steps) | 18 | 21 | 87 | 15 |
| EVA (ours,100 steps) | 20 | 42 | 88 | **0.87** |
| EVA (ours, 500 steps) | **21** | **47** | **88** | 4.6 |

Table 7: The performance of FrameFlow-guidance against sampling steps.

**Image In-painting.** To demonstrate the broader applicability of EVA's geometric method, we conducted an image in-painting case study using EVA algorithms adapted for image tasks. The results

| Hyper-parameter | Solved | mRMSD(%<1) | Des(%) |
|---|---|---|---|
| coupling strength gamma = 0.8 | 17 | **45** | 79 |
| coupling strength gamma = 1.2 | 18 | 38 | **91** |
| starting time t0=0.15 | **20** | 43 | 86 |
| starting time t0=0.05 | 19 | 39 | 88 |
| EVA (original, gamma=1.0, t0=0.1) | **20** | 42 | 88 |

Table 8: More ablation study of EVA.

| | 1YCR | 6EXZ_short | 6E6R_med | 5TRV_short | 3IXT | 6EXZ_long | 1BCF | 2KL8 | 1PRW | 5TRV_med |
|---|---|---|---|---|---|---|---|---|---|---|
| success_rate | 0.67 | 0.33 | 0.33 | 0.33 | 0.08 | 0.08 | 0.58 | 1.00 | 0.33 | 0.25 |
| designable_rate | 0.92 | 1.00 | 1.00 | 1.00 | 0.67 | 0.92 | 0.92 | 1.00 | 0.83 | 1.00 |
| motif_rmsd_rate | 0.75 | 0.42 | 0.42 | 0.50 | 1.00 | 0.08 | 1.00 | 1.00 | 0.50 | 0.33 |
| sc_rmsd_rate | 0.75 | 0.83 | 0.58 | 0.75 | 0.08 | 0.42 | 0.58 | 1.00 | 0.58 | 0.75 |

Table 9: More case study results for the RFDiffusion benchmark.

in Fig. 9 highlight the effectiveness of leveraging spatial context for conditional generation. The modification of EVA for image in-painting task includes: a. The pretrained generative model is changed to the DDPM model pretrained in FFHQ dataset (Chung et al., 2022) b) The rotation alignment is no longer needed since it's a standard 2D image task without any misalignment. Interpolation between masked image and random noise is kept for a better initialization. c) The inference setting and implementation for image in-painting follows DPS (Chung et al., 2022). For the comparison against DPS (Table 10, we report the LPIPS (lower is better) on FFHQ 256 ×256 in different sampling steps (i.e., Neural Function Evaluations, NFE).

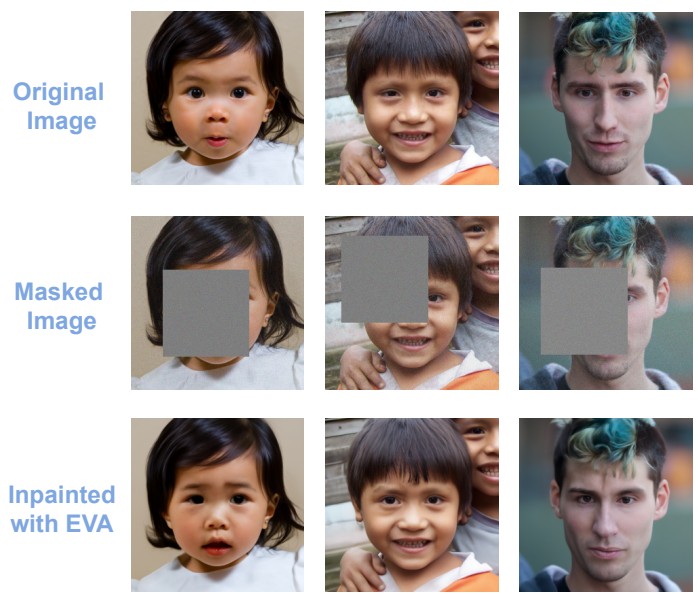

Figure 9: Image in-painting cases of EVA

It is noted that EVA performs better in low NFE setting (NFE≤100), while DPS is better in high NFE setting (DPS's original setting). This aligns with the efficient and direct design of our geometric solution. When the number of sampling steps increases, methods like DPS, which are based on analytical solutions for image in-painting, can better balance the guidance and unconditional scores, thereby achieving superior performance. Since images, unlike proteins, are not 3D geometric entities, making it challenging to fully leverage the advantages of our geometric method. This comparable performance is acceptable and demonstrate our efficiency and effectiveness.

|         | DPS  | EVA  |
|---------|------|------|
| NFE=50  | 0.36 | **0.31** |
| NFE=100 | 0.28 | **0.26** |
| NFE=500 | **0.21** | 0.23 |

Table 10: The performance of EVA in image in-painting. we report the LPIPS (lower is better) on FFHQ 256 ×256 in different sampling steps (i.e., Neural Function Evaluations, NFE).

**Random Error Analysis.** To demonstrate the impact of random errors on the model, we report the success rates and random errors for fixed targets. Under the original inference setting, we generated 100 scaffolds for each target, calculated the number of samples that successfully achieved the motif-scaffolding task, and reported the success rate. We now run this process five times for five targets (with different random seeds as required). Each target yields five success rates, and we report the average success rate and standard deviation (std) for each target as in Table 11.

|                 | 6EXZ     | 6E6R      | 2KL8       | 1BCF     | 1YCR     |
|-----------------|----------|-----------|------------|----------|----------|
| Success Rate(%) | 33±1.0   | 33±0.75   | 100±0.40   | 58±1.5   | 67±1.2   |

Table 11: The success rates and random errors for fixed targets. The headers are the PDB ID of five targets.

The results show that EVA performs consistently on the same case, benefiting from the generation process of the flow-based ODE. As for the ablation study, it is worth noting that a single metric being close does not necessarily mean there is no difference between the two methods. For example, higher designability might result from ignoring the motif constraints. Our method successfully addresses the most motif-scaffolding targets and achieves the best motif reconstruction performance.

**Trajectory Analysis** We provide Trajectory Analysis of FrameFlow-guidance and EVA in Table 12. Empirical results indicate that EVA generates straighter and shorter trajectories.

| Method                                   | t=0.0 (Prior) | t=0.4      | t=0.7      | t=1.0      |
|------------------------------------------|---------------|------------|------------|------------|
| FrameFlow-Guidance (100 inference steps) | 11.1 (7.33)   | 7.2 (5.14) | 4.6 (3.01) | 2.8 (1.74) |
| EVA (100 inference steps)                | 9.3 (6.21)    | 5.6 (3.36) | 2.1 (1.49) | 1.3 (0.82) |

Table 12: Empirical Analysis of Sampling Trajectories for EVA and FrameFlow-Guidance. We calculated the average RMSD between the noisy point clouds and the fully denoised ground truth point clouds for 100 sampling trajectories on the target 1YCR. The values outside the parentheses represent the backbone RMSD, while those inside the parentheses represent the motif RMSD.

## D  IMPACT STATEMENT

While there exists the potential risk that motif-scaffolding methods could be misused to develop harmful drugs, it's important to note that drug development is subject to stringent oversight globally. This rigorous regulatory environment ensures that such misuses can be effectively managed and controlled.

