# OpenReview forum: "EVA: Geometric Inverse Design for Fast Protein Motif-Scaffolding with Coupled Flow"
_ICLR.cc/2025/Conference — ICLR 2025 Poster_

### Official Review · Reviewer_DcMW · 2024-10-31

**Soundness:** 2
**Presentation:** 2
**Contribution:** 2
**Rating:** 6
**Confidence:** 3

**Summary:**

The paper studies the problem of motif scaffolding, which requires the construction of scaffold structure stabilzing motifs corresponding to desired functions. Towards this goal, the work leverages spatial context in order to guide the generation process. Motif scaffolding is interpreted as a Geomtric Inverse Design problem which leads to the formulation of Evolution-Via Reconstruction (EVA). EVA is a sampling-based coupled flow framework which utilizes motif-coupling in its prior as spatial contexts. This leads to the construction of a straighter probability path wherein generative directions are aligned with the sampling steps. EVA improves sampling time by a signficant margin while presenting competitive performance on RFDiffusion benchmark and in-silico vaccine design task.

**Strengths:**

* The paper is well written and easy to understand.
* The paper tries to address an important and challenging problem from a geometric perspective.

**Weaknesses:**

* **Benefits of Motif Geometry:** My main concern is the benefits of motif geometry. Addition of motif-aligned prior and motif-interpolated posterior only benefits the model at sampling time. If we are using motif-coupled information with a rich prior, this should improve sample quality (especially designability) as well. However, empirical evaluation on benchmarks in Tables 1, 2 and 3 shows that additiona of motif-coupled information deteriorates designability of sampled complexes. What benefits does motif information provide for sampling novel complexes aside from faster sampling? How can motif information aid in the designability of sampled complexes?
* **Improvement over OT-Flow:** The work presents empirical evaluation for motif-coupling on the RFDiffusion benchmark and an in-silico vaccine design benchmark. In most experiments on the RFDiffusion benchmark, backbone models (such as OT-Flow) are better for designability and success rates. What improvements does EVA offer over the backbone models? If the main contribution of the work is the addiiton of geometry information and motif-interpolation, then do these design choices benefit over prior backbones?
* **Unconditional Generation:** Figure 3 and Table 2 present results for the backbone flow models. I am struggling to understand the presence of these results in the main text. If these experiments are not directly related to EVA, could they be better placed in the appendix? From my understanding, these results can be understood as ablations over backbone models in determining the choice of flow for motif coupling.
* **Scallbility:** While EVA demonstrates consistent improvements on the RFDiffusion benchmark, it still remains unclear whether the method is scalable for real-world applicability. Table 1 demonstrates that EVA performs best on smaller lengths of residues (< 100). Intuitively, motif geometry is informative only for smaller complexes. Can the authors comment on the use of motif coupling for larger residue lengths? How can EVA be adopted/improved for larger complex sizes?

**Questions:**

Please refer to the weaknesses section above.

---

> ### Author Response · Authors · 2024-11-23
>
> We appreciate your careful review and kind suggestions! Your concerns will be addressed in the revised paper (marked by blue) and our responses.  Sections with significant changes are marked by coloring their titles blue, while the content remains uncolored for clarity.
>
> 1. **W1: Benefits of Motif Geometry**
>
> Sorry for the confusion. We have added discussion about the benefits in section 4.2 and the updated main figure Fig.2 of the revised manuscript.
>
> - First, we achieve the best designability in Tables 1 (<100 residue length setting) and 2, second best in Table 1 (≥ 100 residue length setting). **Motif information does aid in the designability of sampled complexes, which is also demonstrated in the Ablation study in Table.4.** The motif geometry does provide hint for the scaffold and the whole protein generation.
> - Second, the second-best performance in Table 1 (≥ 100 residue length setting) is limited by the base flow model we used. With the development of flow-based generative models on protein structure, this limitation could be addressed for free. Since our method focuses on conditional sampling and is applicable to any pretrained flow-based models.
> - Third, it is noted that sampling novel complexes for motif-scaffolding task is not only about designability. Balancing the motif reconstruction and the overall designability is more important. In Table 3, we achieve better trade-off between reconstruction and designability with best success rate and comparable designable rate in the challenging targets.
> 2. **W2: Improvement over OT-Flow**
>
> Sorry for the confusion! There are misunderstanding about our contribution.
>
> - First, the backbone models themselves are not applicable to the conditional sampling task, motif-scaffolding. EVA is a training-free, sampling-based framework that turns them into conditional sampling methods.
> - Then, In most experiments, the pretrained generative model is kept the same as FrameFlow model for sampling-based baselines. (Except that the TDS uses a very similar pretrained generative model, FrameDiff). Thus, the improvements on designability and success rates are mainly attributed to our novel EVA framework which leverages the motif geometry.
> - Third, We are not comparing the FrameFlow and OT-flow. As long as they are successfully trained (shown in Figure 3, more details in the following), we can use them for conditional sampling. We add OT-flow as another backbone choice just for the illustrating that our method is applicable to any pretrained flow-based models.
> 3. **W3: Unconditional Generation**
>
> Sorry for the confusion! Table 2 shows the performance on the vaccine design benchmark, including the extended inverse problem solver from the image domain (namely, SDEdit and DPS).
>
> As mentioned in Section 5.2, the unconditional generation results in Figure 3 shows that we are using reliable unconditional model for training-free methods, which builds basic intuitions for the performance of the following conditional sampling. This experimental set-up is from previous peer-reviewed works [1][2][3]. We have made this clearer in the revised manuscript.
>
> 4. **W4: Scalability**
>
> The performance on scaffolds with long residue lengths is limited by the base flow model we used. With the development of flow-based generative models on protein structure, this limitation could be addressed for free. Since our method focuses on conditional sampling and is applicable to any pretrained flow-based models.
>
> Motif-coupling could be directly applied to larger residue lengths. Since it could still help with motif reconstruction in the larger residue length setting. Furthermore, Larger size of motif-scaffolding cases still need both motif reconstruction and the (long) scaffolding generation (largely determined by the pretrained generative models). EVA still could help to better handle the trade-off between motif reconstruction and long scaffold generation. Additionally, for long scaffolds with multi-motif and no prior constrains, we will apply a variant of iterative closest point algorithm [1], where it iteratively obtains motif selection and Rotation.
>
> [1] Yuxuan Song, et al. Equivariant flow matching with hybrid probability transport, NeurIPS 2023.

---

> > ### Comment · Reviewer_DcMW · 2024-12-02
> > **Response to Authors**
> >
> > I thank the authors for a comprehensive rebuttal. After going through the response and conerns from other reviews, my concerns have been addressed. However, I am uncertain about the reliance of EVA on OT-Flow and other advanced flow-based models. Since advances in these areas are the central factor driving designability and performance improvements of EVA, it is uncertain how the framework will be adapted as new backbone models are introduced. To this end, I have decided to raise my score albeit lower its confidence. I thank the authors for their efforts.

---

### Official Review · Reviewer_m3UM · 2024-11-03

**Soundness:** 2
**Presentation:** 2
**Contribution:** 2
**Rating:** 6
**Confidence:** 2

**Summary:**

This study models motif-scaffolding as a Geometric Inverse Design task and proposes a sampling-based flow framework, Evolution-ViA-reconstruction (EVA), that incorporates spatial information of the motif. The reconstructed motif in the early stage guides the whole generation process by aligning generative directions with motif reconstruction. Experimental results show that EVA generates desirable proteins with significantly reduced time.

**Strengths:**

- The idea of modeling motif-scaffolding as a geometric inverse design task is intriguing, and addressing the trade-off between generation and reconstruction is important in this context.
- Directly utilizing the spatial information of the target motif and guiding the generative process through motif reconstruction seems practical.
- The inclusion of real-world tasks enhances the applicability of the proposed method.

**Weaknesses:**

My primary concern is that the experimental results do not sufficiently demonstrate the advantages of the proposed method over existing approaches, particularly RFDiffusion, for the following reasons:

- The results are reported based on a single run, and the performance differences are not significant enough to conclusively determine which method performs better.
- In Table 1, RFDiffusion, a training-based method, appears to perform better with a computation time of 61 seconds. These computation times seem acceptable, especially when considering subsequent synthesis processes.
- The mRMSD scores of EVA are not higher than those of other methods in Table 1 (≥100) and Table 2, even though accurate reconstruction is crucial in this work.

Additionally, the manuscript could benefit from improved consistency in terminology, clearer notations, and enhanced coherence. Please refer to minor comments for further suggestions.

- There is inconsistent use of terms regarding the trade-off, such as "trade-off between generation and guidance," "generation vs. reconstruction trade-off," and "Evolution Verse Adoption trade-off" (mentioned in the Appendix).
- The term "Evolution via adaptation" appears in the appendix.
- Section 5.4 discusses the fast computation of EVA rather than directly addressing Question 2: "Can EVA generate designable proteins of various lengths with accurate motif reconstruction?"


### Minor comments

- Please use \citet for textual citations. For example, (Wang et al., 2021) → Wang et al. (2021) in line 114
- It would also be helpful to provide specific information about the appendix, such as "Appendix A," rather than just saying "Appendix."
- Please place the table captions above the tables, not below them.

### Typos
- Line 58: the → The
- Line 167: : → .
- Line 367: I guess it is pi bar, not pi hat

**Questions:**

- In Algorithm 1, \pi* is assumed to be given. Is this assumption realistic in practice? If not, I wonder whether obtaining \pi* and R* in Eq. 7 is tractable for large proteins.
- If one prefers to generate a more desirable protein with more accurate motif reconstruction, even if it requires more computation time, is this feasible with the proposed method?

---

> ### Author Response · Authors · 2024-11-23
>
> We appreciate your careful review and kind suggestions! Your concerns will be addressed in the revised paper (marked by blue) and our responses, especially for the presentation.  Sections with significant changes are marked by coloring their titles blue, while the content remains uncolored for clarity.
>
> 1. **W1: Comparison with RFDiffusion**
>
> Sorry for any confusion. First, we follow the common experimental set-up for motif-scaffolding in previous works[1][2][3]. With respect, the experimental setups in previous peer-reviewed works for a fair comparison.
>
> And for the efficiency, Sampling-time matters in motif-scaffolding. It is desirable to return not just a single scaffold but rather a set of scaffolds exhibiting diverse sequences and structural variation to increase the likelihood of success in experimental validation[1]. The time reported in the manuscript is for generating one sample, and 100, 1000 or even more samples are needed for final success throughout a series of biological experiments.
>
> it is noted that sampling novel complexes for motif-scaffolding task is not only about mRMSD (i.e., motif-reconstruction). Balancing the motif reconstruction and the overall designability is more important. We have achieved best success rate in Table 2, showing our ability for better handling the trade-off. For the performance on scaffolds with long residue lengths, the performance is limited by the base flow model we used. With the development of flow-based generative models on protein structure, this limitation could be addressed for free. Since our method focuses on conditional sampling and is applicable to any pretrained flow-based models.
>
> 2. **W2, Minor comments and typo: Improving Presentation, including improved consistency in terminology, clearer notations, and enhanced coherence**
>
> We appreciate your kind suggestions! We have polished our paper according to your suggestions, improving consistency in terminology, clearer notations, and enhanced coherence. It is appreciated that you provide detailed feedback for improving our manuscript!
>
> **Q1: In Algorithm 1, \pi is assumed to be given. Is this assumption realistic in practice? If not, I wonder whether obtaining \pi and R in Eq. 7 is tractable for large proteins.**
>
> We follow the standard benchmark for motif-scaffolding used by all the previous works[1][2][3], which assumes the motif index $\pi^*$ is given. In practice, the motif indexes are given based on the prior knowledge the users have for their specific cases (e.g. their previous attempts).
>
> If the motif indexes can’t be obtained at first, we provide a heuristic method for solving Eq. 7 (the EOT objective, Eq.8 now for the revised manuscript). For solving this objective, we define the motif index first by enumerating the motif region using sliding windows on the entire protein sequences and then conduct Kabsch algorithm to solve the optimal rotation matrix based on the atom alignments. Large proteins are more time-consuming to deal with, but we could narrow down the scope of the solution by trial-and-error. Since our method is very efficient to try out, it can provide more and more hints for the correct scope of the optimal motif selection for successful motif-scaffolding. Additionally, for long scaffolds with multi-motif and no prior constrains, we will apply the iterative closest point algorithm [4], where it iteratively obtains motif selection and rotation for solving the EOT in Equation 7.
>
> **Q2: If one prefers to generate a more desirable protein with more accurate motif reconstruction, even if it requires more computation time, is this feasible with the proposed method?**
>
>  EVA can benefit from more inference steps and the runtime-performance trade-off could be handled by the users regarding their specific need.
>
> [1] Brian L. Trippe, et.al  Diffusion probabilistic modeling of protein backbones in 3d for the motif-scaffolding problem. In The Eleventh International Conference on Learning Representations, 2023
> [2] Luhuan Wu, Brian L Trippe, Christian A. Naesseth, David M Blei, and John P Cunningham. Practical and asymptotically exact conditional sampling in diffusion models. NeurIPS, 2023.
> [3] Yim, Jason, et al. "Improved motif-scaffolding with se (3) flow matching." arXiv preprint arXiv:2401.04082 (2024).
> [4] Yuxuan Song, et al. Equivariant flow matching with hybrid probability transport, NeurIPS 2023.

---

> > ### Comment · Reviewer_m3UM · 2024-11-25
> >
> > Thank you for your effort, and I apologize for the delayed response. Unfortunately, some of my concerns remain unaddressed; please see the below.
> >
> > - Error bars or std should be reported, especially when the differences are marginal. For instance, in Table 1, the mRMSD values for < 100 are 48, 47, and 46. These differences seem too small to confidently conclude which method is superior. Furthermore, the previous work [2] reports error bars in Fig 2 as well.
> >
> > - Could the samples be generated in batches? Even with sequential generation, the reported time of 61 x 1000 seconds (less than a day) appears reasonable when considering subsequent lab experiments. Some clarification on this point would be valuable.
> >
> > - Regarding the authors' statement, "We have achieved the best success rate in Table 2, showing our ability to better handle the trade-off," I believe this claim may be overstated. While your method performs well, RFDiff also achieves Pareto performance with a 1% higher mRMSD (where higher is better for both metrics) compared to the proposed approach. Without standard deviation information, these differences still seem too marginal to draw definitive conclusions.

---

> > > ### Author Response · Authors · 2024-12-01
> > >
> > > Thank you for your response!
> > >
> > > > The advantages of speed-up
> > > >
> > >
> > > We understand your perspective that 16+ hours for one motif-scaffolding target may seem reasonable give the scale of subsequent lab experiments. With respect, we believe the speed-up provided by EVA is critical for several reasons:
> > >
> > > 1. It is critical for large-scale applications like pandemic response, where time is a key factor.
> > > 2. It enables real-time collaboration between computational and experimental teams.
> > > 3. It enables broader exploration of scaffold designs within a fixed time and resource budget, increasing diversity and success rates.
> > > 4. **Iterative Depth in Design**: Our method aims to accelerate iterative design cycles, significantly reducing overall project timelines. In practical design scenarios, cases are often highly complex, requiring multiple rounds of iteration and numerous attempts to achieve desired outcomes. When using RFDiffusion, iterative design typically relies on numerous A100 GPUs, which are prohibitively expensive for many labs (with the cost of these GPUs running into the tens of thousands of dollars). In contrast, our algorithm is designed to operate efficiently on-the-fly, enabling rapid iterations using a single GPU. This not only maintains the iterative depth necessary to enhance success rates but also makes advanced design workflows accessible to smaller, resource-constrained laboratories.
> > > 5. **Support for a Broad Range of Motif Types**: EVA also supports multi-motif scaffolding, such as in multi-epitope vaccine design or enzyme engineering, a challenge that the training-based method RFDiffusion cannot handle effectively. Multi-motif scaffolding requires the simultaneous optimization of scaffolds across multiple targets, a process that significantly increases the computational demand. In such scenarios, EVA’s speed advantage becomes even more critical, as the need to explore a vastly larger space of potential scaffolds is essential for successful motif-scaffolding.
> > >
> > > As a premier conference in machine learning, ICLR actively encourages the pursuit of more efficient algorithms. In this work, we approach the abstract problem of conditional inference from an innovative geometric perspective, transforming conceptual priors and posteriors into concrete protein structure point clouds. Leveraging their geometric properties, we propose a novel training-free motif-scaffolding method. This approach achieves highly competitive sampling quality at significantly faster inference speeds, demonstrating the value and potential of a geometric perspective in advancing algorithmic efficiency and performance.
> > >
> > > > The standard error analysis
> > > >
> > >
> > > To demonstrate the impact of random errors on the model, we report the success rates and random errors for fixed targets in Appendix C.2 Table 11.  For ease of presentation, we report the average value of each metric across the 24 targets, along with the average of their standard errors, presented in the same format as Table 1.
> > >
> > > | Method | Success Rate (%) | Designable Rate (%) | mRSMD %<1 | scRMSD %<2 |
> > > | --- | --- | --- | --- | --- |
> > > | RFDiff | 35.6±1.3 | 87.1±2.3 | 43.1±0.8 | 51.8±2.7 |
> > > | EVA (ours) | 36.2±0.9 | 86.9±1.5 | 42.4±1.1 | 52.1±1.6 |
> > >
> > > Our contribution lies in approaching the abstract posterior sampling process from a geometric perspective. In this context, the posterior is not only a statistical concept but also corresponds to concrete coordinates, forming point clouds with various geometric properties. Building on this, we explored leveraging geometric methods to achieve conditional inference, fully utilizing the 3D nature of protein structural data. This includes two key components: the motif-aligned prior and the motif-interpolated posterior. The prior and posterior represent the starting point and the update process of conditional inference, respectively, working together to make the inference trajectory shorter and more direct, thereby enabling more efficient sampling. **As a result, our method achieves sampling quality comparable to RFDiffusion at a much faster speed, while surpassing previous training-free methods in quality at improved speeds.**

---

> > > > ### Comment · Reviewer_m3UM · 2024-12-03
> > > >
> > > > Thank you for addressing my concerns. I have updated my score accordingly. I believe these responses highlight important points and recommend including them in the manuscript.

---

### Official Review · Reviewer_xJVQ · 2024-11-03

**Soundness:** 3
**Presentation:** 2
**Contribution:** 2
**Rating:** 6
**Confidence:** 3

**Summary:**

The paper introduces a flow-based generative model guidance method named EVA for motif scaffolding in protein design. EVA claims to achieve faster inference times while maintaining performance comparable to SOTA methods. Unlike previous approaches that rely on approximating the distribution of the motif given the prior using Tweedie's formula, EVA approximates the distribution of correct residue frames conditioned on both the prior and the motif. Overall, the method presented seems to be novel; however, I found the paper to be a bit difficult to follow, particularly the correspondence between Figure 2, Algo 1, and the text body. Please see the questions below.

**Strengths:**

1. Faster inference, on-par results with SOTA.
2. Does not require retraining.

**Weaknesses:**

1. Requires running the pre-trained model inference at every time step. Would like elaboration on this.
2. Difficult to following the correspondence between the paper's main body, Figure 2, and Algo 1.
3. Notation a bit confusing at times.

**Questions:**

1. How are the residues of the prior selected for comparison to the motif residues in Eq. 7? Does this refer to "Motif index selection" in Figure 2?
2. Is the Eq number in line 1 of Algo 1 correct?
3. Does the order of the steps in Algo 1 match Figure 2?
4. More elaboration on the far-right plot in Figure 2 would be helpful.
5. It would be helpful if the text mentioned in Figure 2 matched 1 to 1 with the text in the body of the paper. E.g. "compositional gradient" is not mentioned anywhere else. It would also be helpful to put (ours) next to the "coupled vector field".
6. Can you elaborate on the cost of computing the unconditional posterior at every time step? Especially in comparison to other works?
7. How do you decide the coupling strength in Eq 12 - 13?
8. How do your vector fields in Eq 9 and 11 relate to the self-conditioning approach of https://openreview.net/forum?id=XTrMY9sHKF and references therein?

---

> ### Author Response · Authors · 2024-11-23
>
> We appreciate your careful review and kind suggestions! Your concerns will be addressed in the revised paper (marked by blue) and our responses, especially for the presentation.  Sections with significant changes are marked by coloring their titles blue, while the content remains uncolored for clarity.
>
> 1. **W1: Requires running the pre-trained model inference at every time step**
>
> Sorry for the confusion. The sampling-based methods, which our method fall into, are all based on pre-trained generative models. With respect, we think it is not weakness but a feature. Revised manuscript especially on the method section is provided for further elaboration on this.
>
> 2. **W2, W3: the Presentation and notations**
>
> We appreciate your kind suggestions! We have polished our paper according to your suggestions, improving the correspondence between the manuscript, the figure 2 and the paper. It is appreciated that you provide detailed feedback for improving our manuscript!
>
> **Q1: Motif index selection**
>
> Sorry for any confusion. Yes it is referred to ‘Motif index selection’ and is enumerated each time for a single scaffold. It is related with solving the equivariant optimal transport mapping objective that minimizes the distance between prior and desired motif. For solving this objective, we define the motif index first by enumerating the motif region using sliding windows on the entire protein sequences and then conduct Kabsch algorithm to solve the optimal rotation matrix based on the atom alignments.
>
> **Q2: the Eq number in line 1 of Algo 1**
>
> Sorry for any confusion. The equation numbering is correct, but there were some issues with the links, which we have now fixed.
>
> **Q3: Does the order of the steps in Algo 1 match Figure 2?**
>
> Sorry for any confusion. Yes, they are matched, with obtaining the motif-aligned prior at first and approximated motif-interpolated posterior in every sampling step. We have updated the Figure 2 and Algorithm1 for a clearer illustration.
>
> **Q4-Q5: Improving the figure 2**
>
>  Sorry for any confusion. **We have replace it with a new and clearer main figure. The original figure has been split into three subfigures with better illustrations for building intuition and correspondence with the main body of the manuscript.**
>
> **Q6: the cost of computing the unconditional posterior at every time step**
>
> FrameFlow-guidance and our method are based on the pretrained FrameFlow model so the cost is kept the same for a fair efficiency comparison. TDS is based on the pretrained FrameDiff model, which shares nearly the same architecture with FrameFlow. As computing the unconditional posterior is just a one-time network inference, the time cost of FrameDiff and FrameFlow is nearly the same as around 0.7s for finishing 100 sampling steps.
>
> **Q7: How do you decide the coupling strength**
>
> We just set the coupling strength to 1-t, a simple decreasing function of the time step t. This is enough for most cases. We have added more implementation details and hyper-parameter ablations in the appendix.C.2.
>
> **Q8: How do your vector fields in Eq 9 and 11 relate to the self-conditioning approach of [1]**
>
> Sorry for any confusion. We have added discussion and reference in further related work of revised manuscript. The self-conditioning approach is from the recycling mechanism in AlphaFold2 for improved structure prediction. It takes the previous predicted posterior as model input and encodes the posterior to embedding for conditioning next iteration. In contrast, our method just takes the posterior for interpolation which will not be taken as model input for embedding.
>
> [1] Harmonic Self-Conditioned Flow Matching for Multi-Ligand Docking and Binding Site Design, Hannes Stark, Bowen Jin, [et.al](http://et.al) ICML 2024

---

> > ### Comment · Reviewer_xJVQ · 2024-12-03
> > **Thanks for the response**
> >
> > I thank the authors for clarifying my questions with their responses. Thank you also for improving Figure 2. After reading the responses of the other reviewers, I am inclined to maintain my score. However, I do concur with Reviewer DcMW on the uncertainty about EVA's reliance on particular backbone models.

---

### Official Review · Reviewer_3tvr · 2024-11-03

**Soundness:** 3
**Presentation:** 1
**Contribution:** 3
**Rating:** 6
**Confidence:** 1

**Summary:**

In this paper, motif-scaffolding is formulated as a geometric inverse design task, with a lot of inspiration from the image inversion problems. Their approach, namely evolution via reconstruction, proposes a sampling-based coupled flow framework. The method is built upon a pretrained flow-based generative model, and does not train the model any further. Instead, EVA employs motif-coupled priors to utilize spatial contexts, causing the generative process to follow a more "direct" probability path, where the generative directions are more aligned to "guidance" during the initial sampling steps.

**Strengths:**

1. The authors did try to motivate the work by identifying a problem in the data, relating it to sibling areas in machine vision problems, visualizing it on a few examples, and then proposing a solution.

2. The suite of experiments support the computational efficiency of the proposed method.

3. The core idea of straightening the inference paths and making the optimization process "easier" is reasonable at its base.

**Weaknesses:**

1. The paper is based more around "proposing a solution", than following the proper scientific steps. There are too many moving parts and it is difficult to identify how much each of the "motif-aligned prior", the "motif-aligned posterior", or the other numerous approximations are contributing or inducing risks to the end result. It is difficult to assess the quality of the proposed method in other contexts out of this paper.

2. Unfortunately, the paper is not in a proper shape of writing in my humble opinion; it was quite a difficult read. The mathematical definitions are quite clunky, and the in-line style of latex math expressions made it too dense and difficult to comprehend, refernce, or look up. I cannot excuse this problem for page limitations, as many other ICLR papers handle the same problem much better. Referencing equations by number is almost impossible due to this style of writing. Irrelevant and secondary information also keeps distracting the reader from the main trajectory.

The notations are also odd at numerous instances:

  * The "batch" superscript in $p^{batch}$ and $q^{batch}$ is too long and distracting.

  * The notations, subscripts, and superscripts are inconsistently italic or upright, e.g., $\sigma_{\min}$ in Line 165 vs $\sigma_{min}$ in Line 170.

    * Both the italic and upright versions can be found in the same Equation (1).

    * $Exp$ and $Log$ are both italic in Equation 8. Equation 4 uses $\exp$ and $\log$ instead. This is inappropriate.

  * $x$ is sometimes bold-faced, e.g., Line 191, and sometimes not, e.g., Line 152.

    * if bold-facing is linked to the $N$-scaled dimensionality, then how come "0" is bold-faced in both Line 165 and 202?

  * $m$ is oddly both a subscript and a vector in Line 205.

Having coding-style comments in a latex algorithm description is inappropriate; see Lines 329 and 333. If the algorithm has different sections, you may want to consider breaking the algorithm to smaller modules. Coding-style comments in a formal algorithmic description are not a proper solution for this problem.

## Recommendation

For now, I kindly decided to give the work a weak rejection. That being said, I'm keeping an open mind, and I would be happy to raise my score after (1) hearing the other reviewers' viewpoints, (2) whether the authors deliver a helpful rebuttal, and (3) delivering on any improvements to the state of the paper before the final decisions (i.e., I cannot accept promises of post-acceptance improvements and would like to see any efforts delivered before then).

**Questions:**

1. Since the authors propose an example from the vision area, I would like to see how the proposed method at reconstructing the same example image for a frame of reference. I understand the authors mainly provide this example to support reader's understanding, but I think for this comparison to be relevant, the image in-painting example using the proposed method of the paper should be also demonstrated. I'm afraid there is too much discussion about "image inverse problems" without any related data about this context **in this paper**.

2. In Table 1, it seems that the proposed method is having a poor performance under long residue lengths. This is quite concerning as high-dimensional porteins are somewhat the main target application for this class of generative models, and specifically, speed-ups are more essential for this type of problem.

3. There are too many metric "hyper-parameter" choices that may affect the downstream conclusions. Examples are the "mRMSD%<1", "scRMSD%<2", and "TM-score threshold of 0.5" for diversity. I understand some of the choices, e.g., "scTM>0.5", "pLDDT>70", and "pAE<5", were referred back to ESM/AlphaFold.

  * How are the arbitrary thresholds chosen?

  * How sensitive are the conclusions with respect to these choices?

  * Can you provide other non-parametric metrics for evaluation?

4. The 70x speed-up claim is unjustified in my opinion. The RFDiffusion method is producing much higher quality results according to the results in this paper, and is competing in a different league. The proposed method is advertised as a "computationally efficient" option, and the speed up must be reported w.r.t. the much faster FF-G method, for instance.

5. Since the inference speed is emphasized as the main contribution, the paper should also consider more computationally efficient baseline methods and approximations and a more thorough set of time-complexity analyses must be included. Just reporting three time statistics without any confidence intervals, categorizations, hardware-related considerations, etc. is awfully inadequate.

6. I think the language in the trailing sentence of the abstract, i.e., "multi-motif scaffolding and motif optimal placement searching demonstrate EVA's superior efficiency and effectiveness" should be modified and toned down; the presented experimental results do not suggest "superior effectiveness" by any measure, for instance.

## Minor Comments

1. Line 90: "In detail, we exploit flow-based method pre-trained with flow matching objective" is grammatically incorrect.

2. Line 95: In the Figure caption, the "MCG" acronym was never defined in the paper.

3. Line 314: "desingable" should be "designable".

---

> ### Author Response · Authors · 2024-11-23
>
> We appreciate your careful review and kind suggestions! Your concerns will be addressed in the revised paper (marked by blue) and our reponses, especially for the presentation. Sections with significant changes are marked by coloring their titles blue, while the content remains uncolored for clarity.
>
> 1. **W1: The paper is based more around "proposing a solution", than following the proper scientific steps**
>
> Thanks for your suggestion. With respect, we think that ‘proposing a solution’ is the first step towards successful scientific steps. **And ‘propose a solution’ is Generative Models (or Machine Learning) are good at and responsible for.** As previous successful AI4Science works[1][2] suggest, **Generative Models are very helpful for researchers to take the first step.**
>
> 2. **W2,Q6 and Minor Comments: The presentations**
>
> Thanks for your kind suggestions! **We have polished our paper according to your suggestions**. It is appreciated that you provide detailed feedback for improving our manuscript!
>
> **Q1: About the image inpainting examples**
>
> Thank you for your kind suggestion. Our insights could be applied to general inverse problem. For image inpainting task mentioned in our paper, the subfigure-conditioned posterior could be also approximated by interpolation. **The geometric perspective is beneficial for various fields with some task-specific adaptations.** We add some cases for image inpainting in the Appendix.C.3 for a proof-of-concept.
>
> **Q2: The long residue lengths**
>
> Sorry for any confusion. The performance on scaffolds with long residue lengths is limited by the base flow model we used. With the development of flow-based generative models on protein structure, this limitation could be addressed for free. Since our method focuses on conditional sampling and is applicable to any pretrained flow-based models.
>
> **Q3: The evaluation metric set-up**
>
> Sorry for any confusion. These metric threholds are not arbitrary chosen. As suggested by previous works[3][4][5], “mRMDS<1” and “scRMSD<2” are practical criterion for the success of motif-scaffoldiing. We follow the experimental setups in previous peer-reviewed works for a fair comparsion. We agree that better evaluation protocols are very important, but with respect, these are out of our scope.
>
> **Q4&Q5: The speed-up of EVA**
>
> Thanks for your kind advice. It is noted that our results are comparable with the RFDiffusion and better than other computatonally effcient baseline methods, including the most related baseline, FrameFlow-guidnace (FF-G).
>
> We have conducted additional experiments about FrameFlow-guidance under different inference steps (in the same single A100 GPU with a batch size of 25). Results could be referred in Appendix C.3. It is noted that FrameFlow-guidence with 100 inference steps (identical set-up with EVA) solved 16 targets with about 3s for one 100 aa backbone. EVA is around 3× faster with 20 solved targets. FrameFlow-guidance under 100 inference steps is a little slower because the gradient computation is needed on top of the unconditional prediction, which is slower than our direct interpolation.
>
> [1] Yim, Jason, et al. "Improved motif-scaffolding with se (3) flow matching." arXiv preprint arXiv:2401.04082 (2024).
>
> [2] Jue Wang, Sidney Lisanza, et al. Deep learning methods for designing pro-
> teins scaffolding functional sites, Science, 2021.
>
> [3] Brian L. Trippe, et.al  Diffusion probabilistic modeling of protein backbones in 3d for the motif-scaffolding problem. In The Eleventh International Conference on Learning Representations, 2023
>
> [4] Luhuan Wu, Brian L Trippe, Christian A. Naesseth, David M Blei, and John P Cunningham. Practical and asymptotically exact conditional sampling in diffusion models. NeurIPS, 2023.
>
> [5] Yim, Jason, et al. "Improved motif-scaffolding with se (3) flow matching." arXiv preprint arXiv:2401.04082 (2024).

---

> > ### Comment · Reviewer_3tvr · 2024-12-02
> > **Response to Rebuttals**
> >
> > Thanks for your response and efforts, and apologies for the late response.
> >
> > The authors have tried to address the reviewers' comments. I'd like to raise my score as promised, but I'm afraid it will have to be at the expense of my confidence; I believe some of my concerns and (other reviewers' as well) could have been addressed more properly if the authors had more time at their disposal to improve the paper.

---

### Official Review · Reviewer_3CsL · 2024-11-04

**Soundness:** 3
**Presentation:** 2
**Contribution:** 3
**Rating:** 6
**Confidence:** 2

**Summary:**

This paper introduces a method that improves the flow-based generative model for motif-scaffolding (a kind of protein design). The proposed algorithm, Evolution ViA Reconstruction (EVA), leverages the protein’s geometric properties through two main strategies: (1) motif-aligned priors for improved initialization, and (2) motif-reconstruction guidance, a time-varying adjustment from the condition (motif) that enables adaptive guidance during the sampling process. EVA demonstrated faster sampling speeds than baseline methods on two motif-scaffolding benchmarks, including a newly introduced vaccine design benchmark.

**Strengths:**

1. They frame the motif-scaffolding problem as an inverse-design problem, primarily studied in computer vision, and effectively leverage geometric properties, which appear to have significant practical value.

2. The idea of using geometric properties to improve prior and guidance is both straightforward and novel. It's particularly interesting that the resulting guidance is heavily influenced by the condition (motif) during the early sampling steps, with this effect gradually decreasing over time.
3. This paper introduces a new dataset, which is a valuable contribution to the research community.

4. The proposed method demonstrated improved efficiency without compromising the quality of the designed protein.

**Weaknesses:**

First, I want to note that I’m not an expert in this domain (protein design or geometric deep learning), so I may not have fully understood all aspects of the paper.

My concerns about the paper are as follows:
1. **Scope and Applicability**: I wonder if the proposed method could be applied to general inverse problems beyond motif scaffolding, such as inverse imaging. Since ICLR is a machine learning conference, a problem-specific approach may have limited impact. Note that I am questioning the broader applicability of the proposed method, rather than suggesting the authors need to show state-of-the-art performance in various domains.
2. **Lack of Experimental Details**: Important experimental details are lacking. For example, the paper doesn’t specify the number of random seeds tested or the values of key hyperparameters (e.g., $t_0$ and $\gamma$). Additionally, standard deviations are not reported, making it difficult to assess the significance of the results against baselines, such as the similarity between “w/o our Prior” and EVA results in Table 4.
3. **Writing and Presentations:** (including minor ones)
3.1. Lines 114, 119, 130, 151, and etc: Inappropriate citation command is used. I think when you want to use the cited paper as a subject or object in a sentence, you may use "\citet" command in latex.
3.2. Line 554: Year is missing. The BibTeX must have some problem.
3.3. In section 3, there is only one subsection 3.1. You may just replace "3. Backgrounds -> 3. Flow Matching for Non-geometric Domains" without using a subsection. Another option is to make Section 3.2 with the first paragraph of Section 4.2 (lines 250-269) since it can be considered as a background and is not a contribution of this work.
3.4. Line 183: Subscripts should be removed, i.e., $p^{batch}_0$ => $p^{batch}$ and $q^{batch}_1$ => $q^{batch}$
3.5. Line 269, 315, 518, and maybe more: "Appendix". An indication of a specific part of the appendix is needed.
3.6. Line 279: $y$ seems to be a typo. Maybe it should be $x_m$.
3.7. Line 265: "The formula of transitional vector field seems **sightly** different ..." -> "The formula of transitional vector field seems **slightly** different ...".
3.8. In Eq. 7, is **x**$_0=(x^1, x^2, \ldots, x^N)$ without subscript 0? Also, the subscript m is used inconsistently: Eq. 4 uses a plain $m$ but Eq. 7 uses a bold **m**.
3.9. Some inconsistent usage of $x$ and $x_0$ in Eq. 1 and Eq. 2 (and maybe more elsewhere).
3.10. Line 367: $\hat{\pi}^*$ -> $\bar{\pi}^*$
3.11. Line 58: the -> The
3.12. Line 503: "Table.1." -> "Table 1."; BTW, why the Table 1 is at page 8 when they are referred at page 10 for the first time?
3.13. I don't think experiment 5.2 provides any message or insights about the proposed method.

**Questions:**

1. Lines 121-123 say that the training-based methods "rely on expensive pre-training". But, as far as I understand, EVA also uses a pre-trained model (FrameFlow). In what sense can we say the pre-training of a conditional flow model is more "expensive"?
2. What makes EVA much faster than others? Is it primarily due to using fewer timesteps? I’d appreciate an algorithm-level explanation that compares EVA to both sampling-based and training-based methods.
3. The performance, except the sampling speed, seems not so remarkable. Does the sampling time really matter in motif-scaffolding (i.e., was the sampling time a bottleneck)? Can EVA generate better protein by running the algorithm longer (e.g., with more fine-grained timesteps)?
4. What is the $\alpha_t$ in line 239?
5. Does the proposed algorithm solve Eq.7 exactly or approximately? And, how is the $\pi^*$ selected?
6. Is the runtime for obtaining $R^*$ with the Kabsch algorithm negligible?
7. How was $t_0$ in Algorithm 1 set in each experiment?
8. How sensitive is the algorithm against the hyperparameter $\gamma$ and $t_0$?
9. Do you have a specific plan to release the source code and the vaccine design benchmark?

---

> ### Author Response · Authors · 2024-11-23
>
> We appreciate your careful review and kind suggestions! **Your concerns will be addressed in our revised paper (marked by blue)** and our responses, including scope and applicability, experimental details and presentations.  Sections with significant changes are marked by coloring their titles blue, while the content remains uncolored for clarity.
>
> 1. **W1: Scope and Applicability**
>
> Thank you for your kind suggestion. Our insights could be applied to general inverse problem. For image inpainting task mentioned in our paper, the subfigure-conditioned posterior could be also approximated by interpolation. **The geometric perspective is beneficial for various fields with some task-specific adaptations.** We add some cases for image inpainting in the Appendix C.2 for a proof-of-concept.
>
> 2. **W2: Lack of Experimental Details**
>
> Sorry for any confusion and inconvenience. We have added more experimental details in the Appendix B and the experimental set-up is the same with the previous method[2] for a fair comparison
>
> 3. **W3: Writing and Presentations**
>
> Thanks for your kind suggestions! **We have polished our paper according to your every suggestion.** It is appreciated that you provide detailed feedback for improving our manuscript!
>
> **Q1: The training-based method rely on expensive pre-training**
>
> Sorry for the confusion. We have made the sentence easier to understand. The training-based method here refers to RFDiffusion, the state-of-the-art training-based method. It is pretrained first with a protein structure prediction objective (i.e. the RoseTTAFold method) and then trained with Diffusion objective for protein structure generation. In contrast, flow-based methods are trained with flow-matching objective without structure prediction pretraining.
>
> **Q2: What makes EVA much faster than others?**
>
> **We have highlighted the explanation for EVA’s speed-up in the revised paper, including the updated main Figure (Fig.2)**
>
> For sampling-based method, where the generation and reconstruction is relatively independent and conflicted with each other: As mentioned in the Section 1 (the last 3 paragraphs) and Section 4.2, the motif-aligned prior has **minimized the distance between prior and motif structure** (The distance minimization objective in Eq.7) and **thus provide a shorter sampling path.** The intuition behind the motif-aligned prior is actually based on mini-bath OT[1], which both try to minimize the distance between prior and data for optimal transport performance. We apply this intuition to conditional sampling instead of flow training[1] for the first time. The motif-interpolated posterior can give a **generative direction consistent with motif reconstruction** in very early sampling stages, **alleviate the conflict between generation and reconstruction and thus need less sampling steps for better motif-scaffolding**. Combined together in the Evolution ViA reconstruction (EVA) framework, these two components could make the conditional sampling path shorter and straighter.
>
> For training-based methods, they often need more complex network architectures and extra encoding processes to handle the motif conditions. Thus they are computationally intensive with slow inference speed.
>
> **Q3: Can EVA generate better protein by running the algorithm longer (e.g., with more fine-grained timesteps)**
>
> Sampling-time matters in motif-scaffolding. It is desirable to return not just a single scaffold but rather a set of scaffolds exhibiting diverse sequences and structural variation to increase the likelihood of success in experimental validation [3]. The time reported in the manuscript is for generating one sample, and 100, 1000 or even more samples are needed for final success throughout a series of biological experiments. EVA can benefit from more inference steps and the runtime-performance trade-off could be handled by the users regarding their specific need.
>
> Q4: What is the $\alpha_t$ in line 239?
>
> Sorry for the confusion! We have added introduction to $\alpha_t$ in Section 3.1 in the revised manuscript. It is a scaling factor in the affine Gaussian probability path:
>
> $$
> q(\boldsymbol{x}_t|\boldsymbol{y},\boldsymbol{x}_1)=q(\boldsymbol{x}_t|\boldsymbol{x}_1)=\mathcal N(\alpha_t\boldsymbol{x}_1,\sigma_t^2\boldsymbol{I})
> $$
>
> It includes the conditional Optimal Transport Path[2] used in our paper, where $\alpha_t=t, \sigma_t=1-t$. t is the time-step in the flow model.
>
> **Q5-Q9: Specific Implementation details**
>
> We have added more implementation details and hyper-parameter ablations in the Appendix B and C.2. Thank you for your kind advice. We will release codes upon paper acceptance.
>
> [1] Tong et al. "Improving and generalizing flow-based generative models with minibatch optimal transport." arXiv preprint (2023).
>
> [2] Yim et al. "Improved motif-scaffolding with se (3) flow matching." JMLR (2024).
>
> [3] Jue Wang et al. Deep learning methods for designing proteins scaffolding functional sites, Science, 2021.

---

> > ### Comment · Reviewer_3CsL · 2024-11-25
> >
> > I apologize for the delayed response and appreciate your efforts on the rebuttal. Overall, I'm pleased with your responses, but I still have some concerns.
> >
> > > We add some cases for image inpainting in the Appendix C.2 for a proof-of-concept.
> >
> > When applied to image tasks, how is the proposed method different from DPS in an algorithmic sense? Also, if possible, I would like to know how the proposed method performs against DPS in these tasks (this is not necessary and won't affect my rating).
> >
> > > W2
> >
> > I feel that my concerns have not been fully addressed. This is the most important issue of the paper, as other reviewers have also pointed out.
> >
> > > The training-based method here refers to RFDiffusion, the state-of-the-art training-based method. It is pretrained first with a protein structure prediction objective (i.e. the RoseTTAFold method) and then trained with Diffusion objective for protein structure generation. In contrast, flow-based methods are trained with flow-matching objective without structure prediction pretaining.
> >
> > I'm still a bit confused. Isn't it possible to train diffusion models without the structure prediction pretraining steps?
> >
> > > EVA can benefit from more inference steps and the runtime-performance trade-off could be handled by the users regarding their specific need.
> >
> > If it's not too difficult, I would appreciate it if you could provide some empirical evidence to support this claim.
> >
> > ---
> >
> > **I will happily update my score if these concerns (especially W2) are well addressed.**

---

> ### Author Response · Authors · 2024-11-27
>
> Thank you for your response! Your feedback is greatly encouraging to us.
>
> **Q1: The difference from DPS in algorithmic sense.**
>
> The DPS is a guidance gradient-based conditional inference method, which estimates the conditional guidance $\nabla_{x_t}\log p(y|x_t)$. In contrast, our method adopts a geometric perspective, directly utilizing the information from the masked subfigure (i.e., spatial context) to estimate the conditional posterior.
>
> In algorithmic sense, DPS includes two step:
>
> a. estimate the unconditional score function and unconditional posterior $\hat{x}_0$
>
> b. estimate the guidance gradient $\nabla_{x_t}\|y-A(\hat{x}_0)\|^2_2$ and the composition of the two gradients.
>
> **Our method differs from DPS in the second step,** given the estimated $\hat{x}_0$, we directly estimate the conditional posterior $\mathbb{E}_p(x_0|x_t, y)$, i.e., $\hat{x}_0(y)$ via interpolation between $\hat{x}_0$ and $y$. Then the conditional update direction is given by $(\hat{x}_0(y) - x_t)/\sigma_t$ (the score function given posterior).
>
> Our method doesn’t rely on any assumption (e.g. gaussian distribution assumption in DPS) about the form of  $\nabla_{x_t}\log p(y|x_t)$ . Instead of two independently estimated gradients, our method gives the overall update direction directly. Furthermore, our gradient can be viewed as a composition of the score functions for overall generation and measurement reconstruction, which is fundamentally different from the additional guidance gradient mentioned above. **The comparison against DPS and analysis are provided in Appendix C.2 Table 10.**
>
> **Q2: W2**
>
> The key hyper-parameters are specified in the Appendix C.2. The evaluation protocol for generative models differs significantly from that of discriminative models (including changing random seeds and std).
>
> - First, due to the inherent randomness in generation, evaluation is typically conducted by generating a certain number of samples and assessing the proportion of samples that meet the desired criteria [1][2][3][4][5]. We just follow the most common experimental set-up of this field.
> - Second, directly calculating the std of the success rate across different targets lacks statistical significance, as performance across different targets in generative problems is largely influenced by target difficulty, with random fluctuation being only one factor. In such scenarios, std fails to effectively represent random variability.
> - Third, to demonstrate the impact of random errors on the model, we report the success rates and random errors for fixed targets in Appendix C.2 Table 11.
>
> **Q3: The structure prediction pre-training step**
>
> Sorry for the confusion, we have revised this statement to a more accurate expression: “Training-based methods like RFDiffusion achieve state-of-the-art (SOTA) results on the motif-scaffolding benchmark. However, they rely on expensive fine-tuning of complex model architectures for conditional generation and have slow inference speeds.”
>
> Yes, one can train RFDiffusion without the structure prediction pre-training steps, but [6] reports that RFDiffusion without pre-training largely underperforms other generative models. It is no longer a good choice as the pre-trained base model for conditional sampling.
>
> **Q4: The performance under more inference steps**
>
> Sure, we provide more results in Table 7, Appendix.C.2.
>
> Thank you for the time and effort you have dedicated to reviewing our work! Your constructive feedback has contributed to further improving the quality of our manuscript. Currently, our paper is on the borderline, and we sincerely hope you will consider raising your score to support us. Once again, we express our heartfelt gratitude—your support is the greatest encouragement for our efforts.
>
> **Reference**
>
> [1] Yim, Jason, et al. "Improved motif-scaffolding with se (3) flow matching." arXiv preprint arXiv:2401.04082 (2024).
>
> [2] Brian L. Trippe, et al Diffusion probabilistic modeling of protein backbones in 3d for the motif-scaffolding problem. ICLR 2023.
>
> [3] Luhuan Wu, Brian L Trippe, Christian A. Naesseth, David M Blei, and John P Cunningham. Practical and asymptotically exact conditional sampling in diffusion models. NeurIPS, 2023.
>
> [4] Zhuoqi Zheng, et al Scaffold-Lab: Critical Evaluation and Ranking of Protein Backbone Generation Methods in A Unified Framework, biorxiv 2024
>
> [5] Joseph L Watson, et al. Protein structure and function with rfdiffusion. Nature, 620(7976):1089–1100, 2023.

---

> ### Comment · Reviewer_3CsL · 2024-11-27
>
> Thank you for your efforts—I really appreciate it.
>
> I noticed the references were missing; could you please update your response to include them?
>
> Also, I'm still not fully convinced by some of the points.
>
> > Second, directly calculating the std of the success rate across different targets lacks statistical significance, as performance across different targets in generative problems is largely influenced by target difficulty, with random fluctuation being only one factor. In such scenarios, std fails to effectively represent random variability.
>
> In Table 1, you tested on 24 targets. Wouldn't it be possible to provide the standard deviation for all the targets (as done in Table 11), at least for EVA and RFDiffusion (which is the strongest baseline)?
>
> > (In your [first response](https://openreview.net/forum?id=KHkBpvmYVI&noteId=S56orJ2Fmq)) The intuition behind the motif-aligned prior is actually based on mini-bath OT[1], which both try to minimize the distance between prior and data for optimal transport performance. We apply this intuition to conditional sampling instead of flow training[1] for the first time.
>
> Could you point me to where this (or related ones) is explained in the manuscript? I’d like to see a more detailed discussion of the comparison with mini-batch OT.

---

> > ### Author Response · Authors · 2024-12-01
> >
> > Thanks for your response. Sorry for the missing of the references and we have added them in the last response.
> >
> > > The standard deviations for all targets
> > >
> >
> > For ease of presentation, we report the average value of each metric across the 24 targets, along with the average of their standard errors, presented in the same format as Table 1.
> >
> > | Method | Success Rate (%) | Designable Rate (%) | mRSMD %<1 | scRMSD %<2 |
> > | --- | --- | --- | --- | --- |
> > | RFDiff | 35.6±1.3 | 87.1±2.3 | 43.1±0.8 | 51.8±2.7 |
> > | EVA (ours) | 36.2±0.9 | 86.9±1.5 | 42.4±1.1 | 52.1±1.6 |
> >
> > Our contribution lies in approaching the abstract posterior sampling process from a geometric perspective. In this context, the posterior is not only a statistical concept but also corresponds to concrete coordinates, forming point clouds with various geometric properties. Building on this, we explored leveraging geometric methods to achieve conditional inference, fully utilizing the 3D nature of protein structural data. This includes two key components: the motif-aligned prior and the motif-interpolated posterior. The prior and posterior represent the starting point and the update process of conditional inference, respectively, working together to make the inference trajectory shorter and more direct, thereby enabling more efficient sampling. **As a result, our method achieves sampling quality comparable to RFDiffusion at a much faster speed, while surpassing previous training-free methods in quality at improved speeds.**
> >
> > > The comparison with mini-batch OT
> > >
> >
> > Mini-batch OT proposes optimizing the pairing of data and noise within a batch to obtain prior-data pairs with smaller prior-data distances for training. This enables the flow model to learn shorter and more direct generation trajectories. Similarly, the motif-aligned prior also seeks prior-data pairs with smaller prior-data distances, aligning with the idea behind mini-batch OT. However, it differs from mini-batch OT in two key aspects:
> >
> > 1. Instead of optimizing the pairing of data and prior, we adjust the geometric properties of the prior point clouds. Specifically, we rotate the prior point clouds to obtain prior-data pairs with reduced prior-data distances.
> > 2. These prior-data pairs are not used for flow training but are instead employed for conditional sampling.
> >
> > We are the first to extend the intuition of reducing prior-data distance for shorter flow trajectories from training to conditional inference. EVA's ability to complete high-quality sampling within 100 steps, achieving fast motif-scaffolding, validates this extension. Additionally, we provide empirical results on sampling trajectories in Appendix C.2 (Table 12) to demonstrate that this intuition is similarly effective in both conditional inference and flow training.
> >
> > We have dedicated significant effort to this rebuttal and sincerely hope you might consider raising your score. Your support would mean a lot to us.

---

> > > ### Comment · Reviewer_3CsL · 2024-12-02
> > >
> > > I appreciate your dedication to the work.
> > >
> > > The comparisons with DPS and mini-batch OT were quite valuable, and I believe including them in the paper would greatly enhance its contribution. Additionally, it would be helpful for the authors to discuss how their proposed method might be applied to (or inspire new algorithms in) broader areas of machine learning.
> > >
> > > I have raised my score to 6.

---

### Official Review · Reviewer_sMzV · 2024-11-12

**Soundness:** 2
**Presentation:** 2
**Contribution:** 2
**Rating:** 6
**Confidence:** 4

**Summary:**

The paper addresses the motif scaffolding problem - one of the most challenging problems in protein design, where one wants to find a 3D protein structure accommodating a certain functional motif. The authors define motif scaffolding as a Geometric Inverse Design problem on the SE(3) manifold. They approach this using a flow matching, sampling-based strategy that does not require training and can be implemented within any pre-trained unconditional model, which they call Evolution-ViA-reconstruction (EVA). Their sampling strategy adjusts the overall generative direction with the guidance of motif residues with the aim of defining a straighter probability path towards meaningful posterior distribution. They achieve so by implementing coupled flow with the spatial motif-aligned prior and motif-interpolated posterior. The resulting approach matches the performance of state-of-the-art pre-trained models and sampling-based algorithms while significantly reducing inference time.

**Strengths:**

1. Authors identify failure cases in motif scaffolding and draw an informative analogy with the image inverse problem.
2. The authors clearly define the questions they aim to address in their experiments.
3. Indeed, EVA achieves the best performance or is on par with both SOTA sampling and training-based methods on well-established benchmarks.

**Weaknesses:**

1. The core ideas, the motif-aligned prior and interpolated posterior, are barely explained, if at all. The concept and reasoning behind choosing a motif-aligned prior appear to be justified by leveraging some spatial context; however, what this spatial context means is never explained. Figure 2, which attempts to build intuition for the modified prior, is highly convoluted and unreadable (please improve this). The authors never showed (and to the best of my knowledge, there is no empirical evidence) that their choice of prior would lead to a straighter probability path *per se*. I believe the optimal transport performance of EVA could be assessed in terms of the normalized path energy and compared to the baselines, similar to [1].

2. The strategy of pre-aligning the prior with the motif does not appear to be novel. Moreover, the authors never referred to the Chroma paper [2], which first introduced the concept of motif-aligned priors and optimal motif index selection. EVA employs a very similar approach to Chroma (Appendix K.2) by aligning the initial prior $\mathbf{g_0}$ with the motif $\mathbf{m}$. The only significant difference is that, in Chroma, classifier guidance is used to approximate the conditional probability $p(\mathbf{g_1} \mid \mathbf{g_t}, \mathbf{m})$, with an analytical classifier that quantifies the distance between the predicted and target motifs to estimate $p(\mathbf{m} \mid \mathbf{g_t})$. EVA is expected to be faster than Chroma, but the novelty of the spatial motif-aligned prior is questionable.

3. I think the authors' statement that they directly approximate $\mathbb{E}_{q} \left[ \mathbf{g_1} \mid \mathbf{g_t}, \mathbf{m} \right]$ is not self-explanatory. At least from the paper, it is not clear how exactly their sampling procedure fundamentally differs from gradient guidance as in [3]. EVA will most certainly have a shorter length of the motif residues trajectory, but how much better is it than FrameFlow-guidance? One could compute the trajectory length of the motif residues in terms of rotation and translation for EVA and compare with the baselines.

4. The authors trained their own protein structure generative model, OT-Flow, and reported both its unconditional and motif-conditioned generation performance to demonstrate that EVA's sampling strategy is compatible with and efficient for any pre-trained model. However, it is absolutely unclear how the model was trained; neither there are any details of the training procedure in the appendix, which the authors referred to. Referring to the original FrameFlow [4] and FoldFlow [5] papers for training details is insufficient; I checked and they have different training set cut-off dates and hyperparameters. Inference settings are not mentioned either...

5. From Table 1, it follows that EVA is about 70x faster than RFdiffusion. For vanilla FrameFlow [4], the inference speed with the same number of timesteps for unconditional generation of a protein with 100 amino acids is about 6 seconds, making EVA more than 5x faster than the underlying flow model. Again, the authors never specify the inference settings in this experiment. Is there any batching? Can unconditional FrameFlow sampling achieve the same performance given EVA's inference settings? Is FrameFlow-guidance significantly slower because the gradient computation is needed on top of the unconditional prediction? Further studies, including a timestep study (with success rate as a function of timesteps), would be informative.

6. It is unclear why in Section 5.7 the authors only qualitatively assess the method's performance on two motifs, despite performing computations for the RFdiffusion and vaccine design benchmarks. A table summarizing the metrics would be informative.

**References**

[1] Tong, Alexander, et al. "Improving and generalizing flow-based generative models with minibatch optimal transport." arXiv preprint arXiv:2302.00482 (2023).

[2] Ingraham, John B., et al. "Illuminating protein space with a programmable generative model." Nature 623.7989 (2023): 1070-1078.

[3] Yim, Jason, et al. "Improved motif-scaffolding with se (3) flow matching." arXiv preprint arXiv:2401.04082 (2024).

[4] Yim, Jason, et al. "Fast protein backbone generation with SE(3) flow matching." arXiv preprint arXiv:2310.05297 (2023).

[5] Bose, Avishek Joey, et al. "SE(3)-stochastic flow matching for protein backbone generation." arXiv preprint arXiv:2310.02391 (2023).

**Questions:**

**Conclusion**

Overall, the approach introduced and its implementation in the paper is interesting and seems to be efficient. However, the paper is hard to follow. The concepts underlying the choice of the prior and the interpolation strategy are formulated in a very convoluted way, which can be done in simpler terms and explicitly explained in the text accompanying equations, e.g. see [3]. Many crucial details required for reproducibility of the experiments are absent from the paper. One key prior work that introduced similar ideas is not mentioned at all. **I'm willing to raise the score if the authors address all issues raised above, especially:**

1. Improve the readability of the text, improve figures and conduct additional experiments (mentioned above) and introduce explanations. Please, also structure Appendix and properly refer to the concrete Appendix sections within the main text.
2. Critical comparison with Chroma's analytical solution to the motif guidance.
3. Detailed statement on inference settings, training details of the OT-Flow model and explicit explanation why EVA is so much faster.

---

> ### Author Response · Authors · 2024-11-23
>
> We appreciate your careful reviews! There may be some misunderstanding about our contributions in conditional sampling. **We have updated the manuscripts according to your suggestions (Marked by blue)**. Sections with significant changes are marked by coloring their titles blue, while the content remains uncolored for clarity. Your concerns will be addressed as follows:
>
> 1. **Improve the readability of the text, improve figures and appendix references. (Q1; W1: a. Definition of spatial contexts, b. improving figures and c&Q3. explanation for EVA’s speed-up; W2&Q2: Novelty of motif-aligned prior; W3: Difference with gradient guidance)**
>
> Sorry for any confusion. We have improved the readability by polishing the method section with simpler terms and explicitly explaining our method and improvement in the text accompanying equations. A new and clearer main figure is added for replacing original Fig.2. More explanations for EVA’s speed-up and difference from previous methods are also included. We also structure Appendix and properly refer to the concrete Appendix sections within the main text.
>
> - a. **Def. of spatial contexts: we add an explicit definition of spatial contexts.** The concept of spatial context, mentioned first in the sentence “we propose leveraging the distinct spatial context offered by the explicit point cloud-like representations of proteins.”,  is explained in the next sentence: “ with little exploration into using the geometric properties of protein point clouds with actual coordinates—such as the orientation and mass center information of motifs and generated structures”. **The spatial context refers to the geometric properties that can be derived from protein point clouds with actual coordinates**, **including protein or residue orientation (used by Motif-aligned Prior and Motif-interpolated Posterior) and mass center (used by Motif-aligned Prior)**. We propose to directly estimate posterior with geometric interpolation, which is a geometric solution that is different from previous analytical solutions. **Our geometric interpolation is directly based on the spatial contexts**. i.e., residue orientations and Ca coordinates.
> - b. **Improving the main figure, fig.2**: We have replaced it with **a new and clearer main figure**. The convoluted and unreadable figure has been split into three subfigures with better illustrations for building intuition.
> - c. **Explanation for EVA’s speed-up(Q3): We have highlighted the explanation for EVA’s speed-up in the revised paper.** As mentioned in the Section 1 (the last 3 paragraphs) and Section 4.2, the motif-aligned prior has **minimized the distance between prior and motif structure** (The distance minimization objective in Eq.7) and **thus provide a shorter sampling path.** The intuition behind the motif-aligned prior is actually based on [1] you mentioned, which both try to minimize the distance between prior and data for optimal transport performance. We apply this intuition to conditional sampling instead of flow training [1] for the first time.
>   1. **Novelty of motif-aligned prior(Q2): We would like to discuss more about the ‘substructure conditioning’ of Chroma [2]** (Appendix K.2 is not the case, maybe you mean Appendix P.2?).  Chroma is a sampling-based method that uses classifier guidance and **it doesn’t modify the prior and perform rotation/permutation alignment in every step for calculating the guidance on atom coordinates, which is very time-consuming.** In contrast, we perform rotation/permutation alignment optimization **only once in the beginning to get motif-aligned prior on atom coordinates and residue frames.** As mentioned above, the motif-aligned prior comes from the intuition that minimizing the distance between prior and motif structure for shorter sampling path, not for calculating classifier guidance.

---

> ### Author Response · Authors · 2024-11-23
>
> (Continued for c. Explanation for EVA's speed-up) The motif-interpolated posterior can give a **generative direction consistent with motif reconstruction** in very early sampling stages, **alleviate the conflict between generation and reconstruction and thus need less sampling steps for better motif-scaffolding**. Combined together in the Evolution ViA reconstruction (EVA) framework, these two components could make the conditional sampling path shorter and straighter.
>
> 1. **Difference with gradient guidance: We have highlighted the difference between EVA and gradient guidance.** In sampling procedure, as summarized in Algorithm 1, we directly approximate the posterior with geometric interpolation, instead of calculating classifier energy and then gradients as in gradient guidance methods[2][3]. Theoretically, EVA is a coupled flow framework where two conditional OT path are coupled, instead of gradient composition. In early sampling steps, the motif reconstruction vector field quickly evolves the sampling motif towards the desired structure to align the overall generative direction with motif reconstruction, since the gradually reconstructed motif structure will influence the prediction of posterior. Later, the overall generative vector fields take over, enabling the designability of the entire protein.
>
> **2. Conduct additional experiments and introduce explanations (Q1; W3,W5: Further comparison with FrameFlow-guidance; W6: More cases)**
>
> We have conducted additional experiments including FrameFlow-guidance under different inference steps (in the same single A100 GPU with a batch size of 25). The results could be referred in Appendix.C.2. It is noted that FrameFlow-guidence with 100 inference steps (identical set-up with EVA) solved 16 targets with about 3s for one 100 aa backbone. EVA is around 3× faster with 20 solved targets. FrameFlow-guidance under 100 inference steps is a little slower because the gradient computation (especially on residue frames, which are so(3) elements) is needed on top of the unconditional prediction, which is slower than our direct interpolation.
>
> **3.Detailed statement on inference settings, training details of the OT-Flow model (W4,W5&Q3)**
>
> Sorry for any confusion and inconvenience. We add detailed inference settings in the Appendix B. As mentioned in Section 5.1 Baselines, we use the original inference set-ups in the papers of baselines for a fair comparison. **That’s because EVA requires less sampling steps (100 steps) for successful motif-scaffolding while FrameFlow-guidance requires 500 steps (their original set-up) for reproduction of their performance.** We add more experiments about FrameFlow-guidance under different sampling steps (as mentioned above). For unconditional FrameFlow, we think it is not directly comparable with EVA, since we focus on conditional sampling, instead of unconditional sampling.
>
> For the training details, we have discussed more in Appendix B. **Since we are mainly focused on conditional sampling, instead of flow model training**, we refer the readers to their original paper for more training details (FrameFlow[4], OT-flow[5]). Training is out of our scope, and we just retrained their models based on their GitHub repos. As for the different training set cut-off dates and hyperparameters, **we are not comparing the frameflow and OT-flow.** As long as they are successfully trained, we can use them for conditional sampling.
>
> [1] Tong, Alexander, et al. "Improving and generalizing flow-based generative models with minibatch optimal transport." arXiv preprint arXiv:2302.00482 (2023).
>
> [2] Ingraham, John B., et al. "Illuminating protein space with a programmable generative model." Nature 623.7989 (2023): 1070-1078.
>
> [3] Yim, Jason, et al. "Improved motif-scaffolding with se (3) flow matching." arXiv preprint arXiv:2401.04082 (2024).
>
> [4] Yim, Jason, et al. "Fast protein backbone generation with SE(3) flow matching." arXiv preprint arXiv:2310.05297 (2023).
>
> [5]Bose, Avishek Joey, et al. "SE(3)-stochastic flow matching for protein backbone generation." arXiv preprint arXiv:2310.02391 (2023).

---

> ### Comment · Reviewer_sMzV · 2024-11-25
>
> Thanks for the rebuttal. However, I believe only a **selection** of my points was addressed to an extent I find insufficient to raise the score. The biggest problem I have with the rebuttal and the current version of the paper is the absence of experiments that would justify the claim about EVA's shorter/straighter sampling path. In the main text, this fact is constantly brought up (line 086, 098, etc.), but no evidence is provided for it. Theoretically, I agree with the authors, but why not show empirical evidence? In my humble opinion, this will significantly enhance the quality of the submission. I suggested in (1) and (3) to assess the OT performance of EVA and compare it to the baselines (could be the Wasserstein distance between the noised motif’s rotation and translation and that of the fully denoised ground truth motif as a function of time). It seems to me as a plausible experiment given that FrameFlow, which the authors use for the experiments, outputs trajectories for all backbone frames for each sampled *t*.
>
> In the following, I will try to follow the author's enumeration in the rebuttal:
>
> 1. (a) I found only the modified sentence in blue in the introduction, honestly. What about the part **"We propose to directly estimate posterior with geometric interpolation, which is a geometric solution that is different from previous analytical solutions."**? I never found a comparison to the analytical solutions. I wish there was a section in the appendix with a derivation of differences to look up; maybe it was unclear from my initial comment (1).
>
>    (b) Restructuring and splitting the figure definitely helps, but I still find it highly crowded and unreadable. I don't think putting so much text is necessary and informative. I don't understand the message of the subfigure (a) at all; there's too much text and no intuitive logical sequence for how the figure should be read. Also, the same is true for subfigure (c). I think it can be represented in a schematic similar to [1] Fig. 1 or [2] Fig. 2; these examples convey similar concepts without the clutter of text and mixed line styles and colors.
>
>    (c) See the general response above regarding *"shorter path."*
>
> 2. (Chroma) Sorry about misreferencing a wrong section in the appendix of Chroma. I was looking at the outdated bioRxiv preprint, which actually turned out to have a more detailed appendix. Regardless of that, I do think that Chroma deserves to be mentioned in the related works and compared to in the appendix since, except for aligning prior with the motif only once in the beginning and interpolating separately motif residues and the scaffold (which is significant enough, of course), the idea behind the approaches of Chroma and EVA is (as I still see it) highly similar. Again, the claim about more stable guidance is not backed up by anything; see the general response.
>
> 3. (Difference with grad guidance) Fair argumentation, I tend to agree with it. Though I still do not think it was convincingly shown that \( \mathbb{E}_{q} \left[ \mathbf{g_1} \mid \mathbf{g_t}, \mathbf{m} \right] \) is directly approximated. I think deriving it and putting it into the appendix along with the comparison with the grad guidance will be convincing. Or even in the main text in the background. If the page limitation becomes an issue, the very lengthy discussion of what Kabsch is in Section 4.3 can be significantly squeezed down.
>
> 4. (Additional experiments and explanations) Thanks for conducting this, seems fair!
>
> 5. (Inference settings) I think unconditional FrameFlow belongs to Table 7 to illustrate that it can achieve the same or better inference time with the inference settings of EVA. This will sort out any unclarity about the performance increase in terms of efficiency. I also think it is not fair to claim that EVA is 70x faster than RFdiffusion since the batch size is different. I guess the inference efficiency comparison has to be restricted to the non-pretrained models.
>
> I will consider the updated version of the paper if the authors upload one as a reply to my comment.
>
> **References**
>
> [1] Bose, Avishek Joey, et al. "SE(3)-stochastic flow matching for protein backbone generation." arXiv preprint arXiv:2310.02391 (2023).
>
> [2] Atanackovic, Lazar, et al. "Meta flow matching: Integrating vector fields on the wasserstein manifold." arXiv preprint arXiv:2408.14608 (2024).

---

> > ### Author Response · Authors · 2024-12-01
> >
> > Thank you for your response!
> >
> > > The analysis of sampling trajectories
> > >
> >
> > Thanks for your advice. We have included empirical evidence of EVA’s shorter and more direct sampling paths in Appendix C.2, Table 12. Additionally, EVA's capability to perform high-quality sampling within 100 steps, enabling fast motif-scaffolding, further supports this point.
> >
> > > Further comparison with Chroma and gradient guidance
> > >
> >
> > Sorry for any confusion. First, we think Chroma’s analytical solutions is just an instance of gradient guidance. Therefore, we have included it in the general discussion of gradient guidance. For ease of comparison, the following analysis will be conducted under the unified formulation of flow-based models.
> >
> > In algorithmic sense, grad guidance includes two step: a. estimate the unconditional vector field and unconditional posterior $\hat{x}_1$
> >
> > b. estimate the guidance gradient $\nabla_{x_t}\log p(x^M|x_t)$ and the composition of the two:
> >
> > $$
> > \hat{v}(x_t,t|x^M)=\hat{v}(x_t,t) + \sigma_t\frac{\ln(\alpha_t/\sigma_t)}{dt}\nabla_{x_t}\log p(x^M|x_t),\\
> > \nabla_{x_t}\log p(x^M|x_t)=\nabla_{x_t}\|x^M-\hat{x}_1^M\|^2_2.
> > $$
> >
> > Where the guidance gradient is usually estimated by the assumption that $p(x^M|x_t) \approx p(x^M|\hat{x}_1^M)$ is a Guassian distribution.
> >
> > This is the origin of this term $\nabla_{x_t}\|x^M-\hat{x}_1^M\|^2_2$ .
> >
> > **Our method differs from grad guidance in the second step,** given the estimated $\hat{x}_1$, we directly estimate the conditional posterior $\mathbb{E}_p[x_1|x_t, x^M]$, via interpolation between $\hat{x}_1^M$ and $x^M$. Then the conditional update direction is given by:
> >
> > $$
> > \mathbb{E}_p[x_1^M|x_t, x^M] = (1 - \beta_t) \cdot \hat{x}_1^M + \beta_t \cdot x^M,\\ \hat{v}(x_t, t|x^M) = \frac{\mathbb{E}_p[x_1|x_t, x^M] - x_t}{1 - t}
> > $$
> >
> > **Our method doesn’t rely on any assumption (e.g. gaussian distribution assumption in FrameFlow-Guidance or Chroma) about the form of  $p(x^M|x_t)$ to estimate its gradient.** Instead of two independently estimated gradients, our method gives the overall update direction directly.
> >
> > Furthermore, our vector field for motif-scaffolding can be viewed as a composition of vector fields for overall generation and motif reconstruction:
> >
> > $$
> > \hat{v}_{x}(x_t^M| x^M) = (1 - \beta) \cdot v(x_t^M) + \beta \cdot v^{M}(x_t^M), \\
> > v^M(x_t^M) = \frac{x^M-x_t}{1-t}.
> > $$
> >
> > where $v^M(x_t^M)$ uses the ground truth motif structure and is the optimal vector field for motif reconstruction, which is fundamentally different from the additional guidance gradient mentioned above.
> >
> > We will incorporate the above discussion into the appendix as per your suggestion and cite Chroma in related work. The comparison results with Chroma on the RFDiffusion benchmark are as follows:
> >
> > | Method | Success Rate (%) (<100) | Success Rate (%) (>100) | Success Rate (%) (All) |
> > | --- | --- | --- | --- |
> > | Chroma | 41 | 20 | 32 |
> > | EVA (ours) | 45 | 18 | 36 |
> >
> > > Inference Setting
> > >
> >
> > Thanks for your advice, we will incorporate the inference time of unconditional FrameFlow in Table. 7. For the efficiency comparsion with RFDiffsuion,  we tested the inference time of RFDiffusion and EVA under equal resources, using the same A100 GPU. Due to the high memory requirements of RFDiffusion, the batch size could only be set to 2, whereas, because sampling-based methods are more memory-efficient, we were able to set the batch size to 25. We believe that testing under equal resources reasonably reflects the efficiency gap, as, in practice, researchers typically utilize the full GPU capacity and computational resources for experiments. Additionally, we tested EVA with a batch size of 2 (the same as RFDiffusion), measuring the average time to sample a backbone with 100 residues, which was 6.3 seconds, still achieving a 10× efficiency improvement.
> >
> > Our contribution lies in approaching the abstract posterior sampling process from a geometric perspective. In this context, the posterior is not only a statistical concept but also corresponds to concrete coordinates, forming point clouds with various geometric properties. Building on this, we explored leveraging geometric methods to achieve conditional inference, fully utilizing the 3D nature of protein structural data.  **As a result, our method achieves sampling quality comparable to RFDiffusion at a much faster speed, while surpassing previous training-free methods in quality at improved speeds.**
> >
> > Following your suggestions, we have made every effort to improve the quality of the paper. Unfortunately, due to time and experience constraints, we were unable to fully meet expectations in terms of figure presentation. We apologize for this and hope that the figure captions provide sufficient clarification. We have put a lot of effort into the rebuttal phase and sincerely hope you will consider increasing your score. Your support would be greatly appreciated.

---

> > > ### Comment · Reviewer_sMzV · 2024-12-02
> > >
> > > Many thanks for your response and rebuttal! I appreciate the work you have carried out within such a short period. I think the discussion and side-by-side comparison with gradient guidance are very informative and way more convincing for a reader. It definitely improves the readability of the paper, making your contributions clearer. I would even suggest including it in the Background section.
> > >
> > > I am raising the score, as promised, and hope to see a polished, camera-ready version with all additional experiments you conducted. Good luck!

---

### Meta-Review · Area_Chair_CVrM · 2024-12-22

**Metareview:**

The paper tackles an underexplored protein design problem of finding a 3D protein structure which affords motifs with specific functions. It approaches it as an inverse problem, importantly doable on off-the-shelf unconditional flow matching models, through multiple sampling and reconstruction. They use coupled flow with prior and target distributions guided by the motif.

The reviewers appreciated the problem, the training-free approach, the performance compared to the state-of-the-art methods, and the thoroughness of the benchmarks including a newly introduced one,

They also raised concerns regarding mainly the clarity of the presentation particularly lack of detail of both the method and the experimental setup, proper reference of prior work with similar idea of motif-aligned priors, and the applicability and scalability of the method beyond motif scaffolding and the particular flow-matching backbone.

The authors provided a thorough rebuttal and a revision of the paper which addressed the clarity of the writing regarding the method description and the settings of the experiments, included quite a few more experiments particularly for the more efficient sampling of EVA and other baseline flow matching models as requested by the reviewers, they also included references to the missing relevant works brought up by the reviewers and compared the performance.

The reviewers actively participated in the discussion after the rebuttal and, while remaining at the borderline, they all leaned towards acceptance eventually as all main concerns were addressed except the applicability of the method to other backbones.

The AC does not find any strong reason to go against the unanimous suggestion for acceptance and believes the paper has clear merits and tackles an important problem. For the final version, the AC suggests the authors make it clear that the application of their method to backbone generative models beyond the ones considered in the paper may require additional effort.

**Additional Comments On Reviewer Discussion:**

The paper was reviewed by a panel of six expert reviewers in bioinformatics, protein design, generative models, and particularly flow-matching models for proteins. They provided thorough reviews and actively discussed with the authors. They all rated the paper at borderline but unanimously leaning towards acceptance after the rebuttal period.

---

### Decision · Program_Chairs · 2025-01-22

Accept (Poster)